# ZERO-SHOT ROBUSTIFICATION OF ZERO-SHOT MODELS

**Dyah Adila**[*]**, Changho Shin**[*]**, Linrong Cai, Frederic Sala**
Department of Computer Science
University of Wisconsin-Madison
{adila,cshin23,lcai54,fredsala}@wisc.edu

## ABSTRACT

Zero-shot inference is a powerful paradigm that enables the use of large pretrained models for downstream classification tasks without further training. However, these models are vulnerable to inherited biases that can impact their performance. The traditional solution is fine-tuning, but this undermines the key advantage of pretrained models, which is their ability to be used out-of-the-box. We propose ROBOSHOT, a method that improves the robustness of pretrained model embeddings in a fully zero-shot fashion. First, we use language models (LMs) to obtain useful insights from task descriptions. These insights are embedded and used to remove harmful and boost useful components in embeddings—without any supervision. Theoretically, we provide a simple and tractable model for biases in zero-shot embeddings and give a result characterizing under what conditions our approach can boost performance. Empirically, we evaluate ROBOSHOT on nine image and NLP classification tasks and show an average improvement of 15.98% on worst group accuracy, with trivial decrease in overall accuracy over several zero-shot baselines. Additionally, we demonstrate that ROBOSHOT is compatible with a variety of pretrained and language models and propose a way to further boost performance with a zero-shot adaptation variant.[1]

## 1 INTRODUCTION

Zero-shot prediction is among the most exciting paradigms in machine learning. Zero-shot models obviate the need for data collection and training loops by simply asking for a prediction on any set of classes. Unfortunately, such models inherit biases or undesirable correlations from their large-scale training data (Dixon et al., 2018; Torralba & Efros, 2011). In a now-canonical example (Koh et al., 2021), they often associate `waterbirds` with `water background`. This behavior leads to decreased performance, often exacerbated on rare data slices that break in-distribution correlations.

A growing body of literature (Yang et al., 2023; Goyal et al., 2022; Zhang & Ré, 2022) seeks to improve robustness in zero-shot models. While promising, these works require labeled data to train or fine-tune models, and so **do not tackle the zero-shot setting.** A parallel line of research seeking to debias word embeddings (Aboagye et al.; Bolukbasi et al., 2016; Dev & Phillips, 2019; Lauscher et al., 2020) often sidesteps the need for labeled data. Unfortunately, these works often require domain expertise and painstaking manual specification in order to identify particular concepts that embeddings must be invariant to. As a result, out-of-the-box word embedding debiasing methods also cannot be applied to zero-shot robustification.

Can we robustify zero-shot models without (i) labeled data, (ii) training or fine-tuning, or (iii) manual identification? Surprisingly, despite this seemingly impoverished setting, it is often possible to do so. Our key observation is that language models **contain actionable insights** that can be exploited to improve themselves or other models. These insights are noisy but cheaply available at scale and can be easily translated into means of refinement for zero-shot representations. These refinements improve performance, particularly on underperforming slices, at nearly no cost.

---

[*]These authors contributed equally to this work
[1]Code can be found in https://github.com/SprocketLab/roboshot

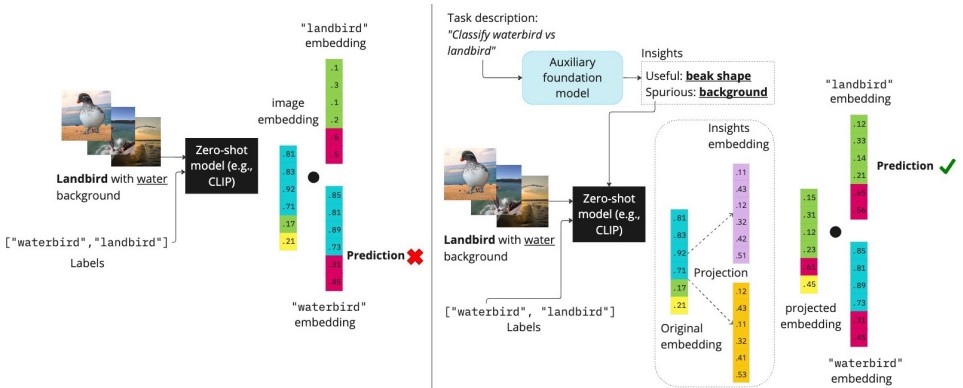

Figure 1: Left: vanilla zero-shot classification. Right: RoboShot projects original embeddings to a space with *reduced spurious components* and *increased useful components*
.

We propose RoboShot, a system that robustifies zero-shot models via language model-based insights *without labels, training, or manual specification*. Using just the task description, RoboShot obtains *positive and negative insights* from a language model (potentially the model to be improved itself). It uses embeddings of these noisy insights to recover *harmful, beneficial*, and *benign* subspaces of zero-shot latent representation spaces. Representations are then modified to neutralize and emphasize their harmful and beneficial components, respectively.

Theoretically, we introduce a simple and tractable model to capture and quantify failures in zero-shot models. We provide a result that characterizes the *quantity and quality* of insights that must be obtained as a function of the severity of harmful correlations. Empirically, RoboShot achieves 15.98% improvement across nine image and NLP datasets while offering sufficient versatility to apply to a diverse variety of base models. Most excitingly, in certain cases, it reaches comparable or greater improvements **even when compared to fine-tuned models** that rely on labeled data. In summary, our contributions include:

1. A simple theoretical model describing zero-shot failures along with a theoretical analysis of our approach that characterizes the amount of information required for obtaining improvements as a function of the most harmful unwanted correlation,

2. RoboShot, an algorithm that implements our core idea. It extracts insights from foundation models and uses them to improve zero-shot representations,

3. Extensive experimental evidence on zero-shot language and multimodal models, showing improved worst-group accuracy of 15.98% across nine image and NLP datasets,

4. A technique to add further robustness by training an adapter *without any labels* requiring only minimal amounts of validation data.

## 2 RELATED WORK

We describe related work in zero-shot model robustness and debiasing embeddings. We provide a more exhaustive list of related work in Appendix B, including papers studying guiding multi-modal models using language and using LMs as prior information.

**Zero-shot inference robustness.** Improving model robustness to unwanted correlations is a heavily studied area (Sagawa et al., 2019; Arjovsky et al., 2019; Krueger et al., 2021; Kirichenko et al., 2022; Liu et al., 2021; Lee et al., 2022). Some methods require training from scratch and are less practical when applied to large pretrained architectures. Existing approaches to improve robustness *post-pretraining* predominantly focus on fine-tuning. (Yang et al., 2023) detects spurious attribute descriptions and fine-tunes using these descriptions. A specialized contrastive loss is used to fine-tune a pretrained architecture in (Goyal et al., 2022) and to train an adapter on the frozen embeddings in (Zhang & Ré, 2022). While promising, fine-tuning recreates traditional machine learning pipelines (e.g., labeling, training, etc.), which sacrifices some of the promise of zero-shot models. In contrast, our goal is to avoid any training and any use of labeled data. Concurrent work seeks to

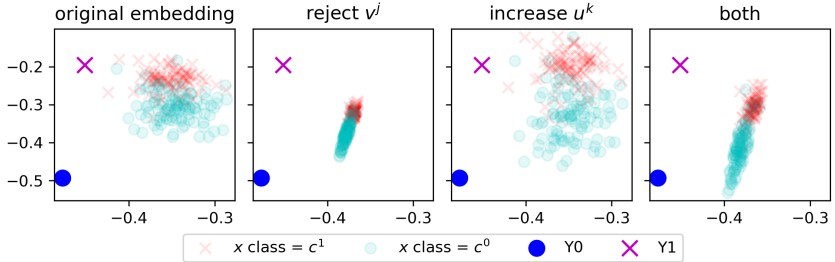

Figure 2: Visualization on CelebA (200 random samples). L-R: (i) original embedding (ii) harmful concept removal (iii) helpful concept addition (iv) full ROBOSHOT. $Y0$ and $Y1$ are class labels

robustify CLIP zero-shot predictions against spurious features by debiasing the classifier (i.e., the labels embedding) against harmful concepts (Chuang et al., 2023)—but does so via manual specification. In contrast, our work amplifies helpful concepts and automates the process of obtaining debiasing vectors.

**Debiasing embeddings.** A parallel line of work seeks to debias text embeddings (Aboagye et al.) (Bolukbasi et al., 2016) (Dev & Phillips, 2019) (Lauscher et al., 2020) and multimodal embeddings (Wang et al., 2022; Berg et al., 2022; Wang et al., 2021) by removing subspaces that contain unwanted concepts. We use a similar procedure as a building block. However, these methods either target specific fixed concepts (such as, for example, gender in fairness contexts) or rely on concept annotations, which limits their applicability across a wide range of tasks. In contrast, our method automates getting *both beneficial and unwanted concepts* solely from the task descriptions. Moreover, our goal is simply to add robustness at low or zero-cost; we do not seek to produce fully-invariant representations as is often desired for word embeddings.

## 3   ROBOSHOT: ROBUSTIFYING ZERO-SHOT MODELS

We are ready to provide our setup and describe the ROBOSHOT algorithm. As mentioned before, we use embedding debiasing principles as building blocks. For our purpose, we utilize concepts obtained from language models and get their embeddings to build the beneficial and unwanted concept subspaces to work with. We call these embeddings the *insight representations*.

### 3.1   MODELING AND SETUP

Suppose that the zero-shot model's latent space contains an (unknown) *concept set*; similar notions have been studied frequently in the literature (Dalvi et al., 2022). For simplicity, we assume that this concept set is given by the orthonormal vectors $\{z_1, \ldots, z_k\}$. The model's encoder produces, for a particular input, a representation $x$ that is a mixture of concepts $\sum_i \gamma_i z_i$, where $\gamma_i \geq 0$ are weights.

We work with the following theoretical model for zero-shot classification. For simplicity, we assume that there are two classes. It is straightforward to extend the analysis below to multi-class. We take $\sum_i \alpha_i z_i$ to be the embedding of a datapoint, while $c^0 = \sum_i \beta_{i,0} z_i$ is the embedding of the first class and $c^1 = \sum_i \beta_{i,1} z_i$ is that of the second. We assume that we have access to $m$ answers $v^1, \ldots, v^m$ from a set of queries to the language model; we describe how these queries are used practically further on. These are given by $v^j = \sum_i \gamma_{i,j} z_i$ for $j \leq m$. We call these *insight representations*.

In the standard approach, the prediction is made by $\hat{y} = \mathbb{1}\{(\sum_i \alpha_i z_i)^T (\sum_i \beta_{i,0} z_i) < (\sum_i \alpha_i z_i)^T (\sum_i \beta_{i,1} z_i)\}$, so that we predict the class that has the higher inner product with the datapoint's embedding. Next, we assume that each input representation $x$ can be represented by partitioning the mixture components into three groups,

$$x = \sum_{s=1}^{S} \alpha_s^{\text{harmful}} z_s + \sum_{r=S+1}^{S+R} \alpha_r^{\text{helpful}} z_r + \sum_{b=S+R+1}^{S+R+B} \alpha_b^{\text{benign}} z_b. \tag{1}$$

In other words, representations comprise of mixture of embeddings pertaining to harmful, helpful, and benign or neutral concepts—this holds for class and insight representations. In Appendix F.5, we empirically show that this assumption holds in real scenarios.

**Example.** We illustrate how harmful correlations produce errors on rare slices of data through a standard task setting, Waterbirds (Koh et al., 2021). Here the goal is to classify `landbirds` versus `waterbirds`, and the background (`land` or `water`) is spurious. Suppose that we have these terms relate to concepts such that $z_{\texttt{water}} = -z_{\texttt{land}}$ and $z_{\texttt{waterbird}} = -z_{\texttt{landbird}}$.

Consider a datapoint coming from a data slice rarely encountered in the training set, for instance, an image of landbird over water. Its embedding might be $x = 0.7z_{\texttt{water}} + 0.3z_{\texttt{landbird}}$. We may also have that $c^{\texttt{waterbird}} = 0.4z_{\texttt{water}} + 0.6z_{\texttt{waterbird}}$ and $c^{\texttt{landbird}} = 0.4z_{\texttt{land}} + 0.6z_{\texttt{landbird}}$. Then, $x^T c^{\texttt{waterbird}} = 0.1 > x^T c^{\texttt{landbird}} = -0.1$, which gives us waterbird prediction, and is incorrect. This is caused by the presence of harmful components in *both* the class embedding (caused by seeing too many images with water described as waterbirds) and the datapoint embedding (where the water background appears). Our goal is to *remove* harmful components (the $z_s$'s) and *boost* helpful components (the $z_r$'s)—without labels or training. Our approach follows.

---

**Algorithm 1: ROBOSHOT**

1: **Parameters:** Input embedding $x$, class embeddings $c^0, c^1$, harmful insight representations $v^1, \ldots, v^S$, helpful insight representations $u^1, \ldots, u^R$
2: **for** $j \in \{1, 2, \ldots, S\}$ **do**
3:     Remove harmful insight $v^j$: set $x \leftarrow x - \langle x, v^j \rangle / \langle v^j, v^j \rangle v^j$
4:     Renormalize $x = x / \|x\|$
5: **end for**
6: **for** $k \in \{1, 2, \ldots, R\}$ **do**
7:     Amplify helpful insight $u_k$: set $x \leftarrow x + \langle x, u^k \rangle / \langle u^k, u^k \rangle u^k$
8: **end for**
9: $\hat{y} = \mathbb{1}\{x^T c^0 < x^T c^1\}$
10: **Returns:** Robustified zero-shot prediction $\hat{y}$

---

## 3.2 ROBOSHOT: ROBUSTIFYING ZERO-SHOT INFERENCE

We describe ROBOSHOT in Algorithm 1. It uses representations of insights from language models to shape input and class embeddings to remove harmful components and boost helpful ones. Figure 2 is helpful in understanding the intuition behind these procedures. Note how unhelpful directions are neutralized while perpendicular directions are boosted.

**Obtaining insight representations from LMs.** The first question is how to obtain insight representations in a zero-shot way– we use *textual* descriptions of harmful and helpful concepts by querying language models using *only the task description*. For example, in the Waterbirds dataset, we use the prompt "What are the biased/spurious differences between waterbirds and landbirds?". We list the details of the prompts used in Appendix D.2. Let $s^1, s^2$ be the text insights obtained from the answer (e.g., {'water background,' 'land background'}). We obtain a spurious insight representation by taking the difference of their embedding $v = (g(s^1) - g(s^2)) / \|g(s^1) - g(s^2)\|$, where $g$ is the text encoder of our model. In addition to attempting to discover harmful correlations, we seek to discover helpful components in order to boost their magnitudes past the harmful ones. We obtain insight representations using language models. For example, we ask "What are the true characteristics of waterbirds and landbirds?' and obtain e.g., {'short beak,' 'long beak'}. The remainder of the procedure is identical to the case of harmful components.

Prompting a language model is typically inexpensive, which will enable obtaining multiple insight vectors $\tilde{v}^1, \ldots, \tilde{v}^m$. From these, we obtain an orthogonal basis $v^1, \ldots, v^m$ separately for harmful and helpful components using standard matrix decomposition methods. Thus we have access to recovered subspaces spanned by such components.

**Removing and boosting components.** ROBOSHOT applies simple vector rejection to mitigate harmful components (lines 2-5 of Algorithm 1) and boosts helpful ones (lines 6-9). To see the impact of doing so, we return to our earlier example. Suppose that we have a single harmful insight $v^{\text{harmful}} = 0.9z_{\texttt{water}} + 0.1z_{\texttt{landbird}}$ and a single helpful insight $v^{\text{helpful}} = 0.1z_{\texttt{water}} + 0.9z_{\texttt{landbird}}$. Note that even these insights can be imperfect: they do not uniquely identify what are harmful or helpful concepts, as they have non-zero weights on other components.

From removing the harmful component (ignoring normalization for ease of calculation), we obtain $\hat{x} \leftarrow x - \frac{\langle x, v^{\text{harmful}}\rangle}{\langle v^{\text{harmful}}, v^{\text{harmful}}\rangle} v^{\text{harmful}} = -0.0244 z_{\texttt{water}} + 0.2195 z_{\texttt{landbird}}$. Then, we already we have that $x^T c^{\texttt{waterbird}} = -0.1415 < x^T c^{\texttt{landbird}} = 0.1415$, thus the correct class is obtained. From a single insight we have neutralized a harmful correlation and corrected what had been an error. Adding in the helpful component further helps. Using vector addition equation in Algorithm 1 line 7, we obtain $-0.0006 z_{\texttt{water}} + 0.4337 z_{\texttt{landbird}}$. This further increases our margin. Note that it is not necessary to fully neutralize (i.e., to be fully invariant to) spurious or harmful components in our embeddings. The only goal is to ensure, as much as possible, that their magnitudes are reduced when compared to helpful components (and to benign components). In Section 4, we provide a theoretical model for the magnitudes of such components and characterize the conditions under which it will be possible to correct zero-shot errors.

### 3.3 LABEL-FREE ADAPTATION (LFA)

Additionally, we explore the limit of neutralizing harmful and boosting helpful insights in the embedding space via *an alternative adaptation approach* when users seek to maximize robustness and have access to *unlabeled* training set and small labeled validation set (with as few as 100 samples). We learn a feature space parameterized by projection matrix $\Pi : \mathbb{R}^d \to \mathbb{R}^d$, where $d$ is embedding dimension. We optimize $\Pi$ so it projects $x$ to a space with minimum dot product with harmful insights $\langle \Pi x, v \rangle$, and maximum with

---

**Algorithm 2:** Label-free adaptation

1: **Parameters:** Input embedding matrix $X$, ROBOSHOT projected embedding matrix $X_{proj}$, spurious insight representations $v$, useful insights representations $u$, class embeddings $c^0, c^1$, epoch number $e$
2: Initialize $\Pi = X_{proj} X^\dagger$
3: **for** epoch in $1, 2, \ldots, e$ **do**
4: $\quad \Pi_{i+1} \leftarrow \arg\min_\Pi \mathbb{E}_x[\mathcal{L}_{LFA}(\Pi_i x, u, v)]$
5: **end for**
6: $\hat{y} = \mathbb{1}\{\Pi x^T c^0 < \Pi x^T c^1\}$
7: **Returns:** Robustified zero-shot prediction $\hat{y}$

---

the helpful ones $\langle \Pi x, u \rangle$. More formally, $\Pi = \arg\min_\Pi \mathbb{E}_x\left[\mathcal{L}_{LFA}(\Pi x, u, v)\right]$. The loss is given by

$$\mathcal{L}_{LFA}(\Pi x, u, v) = \frac{1}{|S|}\sum_{j=1}^S \left\langle \Pi x, v^j \right\rangle - \frac{1}{|R|}\sum_{k=1}^R \left\langle \Pi x, u^k \right\rangle,$$

where $S$ and $R$ are the number of harmful and helpful insights. We observed that the best results are achieved by initializing $\Pi$ as the ROBOSHOT projection matrix, $\Pi_0 = X_{proj} X^\dagger$, where $X = [x_1 \quad x_2 \quad \cdots \quad x_N]$ is the embedding matrix, $X^\dagger$ its Moore-Penrose pseudo-inverse, and $X_{proj}$ is the ROBOSHOT projection matrix. Algorithm 2 details LFA algorithm. We draw inspiration from (Chen et al., 2023) where the authors learn an orthogonal feature space from a source domain dataset and adapt it to a target domain. In contrast to this approach, our focus is on learning the feature space *without any training labels* and using insights as the only form of supervision.

## 4 THEORETICAL ANALYSIS

Next, we provide an analysis that characterizes under what conditions ROBOSHOT can correct zero-shot errors. First, we consider the following error model on the weights of the representations. For all benign representations, we assume $\alpha_b, \beta_b, \gamma_b \sim \mathcal{N}(0, \sigma^2_{\text{benign}})$. That is, the magnitudes of benign components are drawn from a Gaussian distribution. The value of $\sigma_{\text{benign}}$ is a function of the amount of data and the training procedure for the zero-shot model. Appendix F.5 empirically shows that in real scenarios, benign components can be canceled out, indicating that this assumption often holds.

Next, we assume that the insight embedding $v^s = \sum_{i=1}^k \gamma_{i,s} z_i$ (where $1 \le s \le S$) satisfies the property that for $i \ne s$, $\gamma_{i,s} \sim \mathcal{N}(0, \sigma^2_{\text{insight}})$, while $\gamma_{s,s}$ is a constant. In other words, the vectors $v^1, \ldots, v^S$ spanning the harmful component subspace are well-aligned with genuinely harmful concepts, but are also affected by noise. Similarly, we assume that helpful insights $v^r = \sum_{i=1}^k \gamma_{i,r} z_i$ (where $S+1 \le r \le S+R$) satisfy the same property. We seek to understand the interplay between

this noise, benign noise, and the coefficients of the other vectors (i.e., helpful components). Let the result of ROBOSHOT with insight representations $v^1, \ldots, v^{S+R}$ be

$$\hat{x} = x - \sum_{s=1}^{S} \frac{x^T v^s}{||v^s||^2} v^s + \sum_{r=S+1}^{S+R} \frac{x^T v^r}{||v^r||^2} v^r = \sum_{i=1}^{S+R+B} A_i z_i.$$

We first provide a bound on $A_s$, the targeted harmful concept coefficient after applying ROBOSHOT.

**Theorem 4.1.** *Under the noise model described above, the post-ROBOSHOT coefficient for harmful concept $s$ $(1 \leq s \leq S)$ satisfies*

$$|\mathbb{E} A_s| \leq \left| \frac{(k-1)\alpha_s \sigma_{insight}^2}{\gamma_{s,s}^2} \right| + \left| \sum_{t=1, t \neq s}^{S+R} \frac{\alpha_s \sigma_{insight}^2}{\gamma_{t,t}^2} \right|,$$

*where $k$ is the number of concepts ($k = S + R + B$).*

The proof is included in Appendix C.3. The theorem illustrates how and when the rejection component of ROBOSHOT works—it scales down harmful coefficients at a rate inversely proportional to the harmful coefficients of the insight embeddings. As we would hope, when insight embeddings have larger coefficients for harmful vectors (i.e., more precise in specifying non-useful terms), ROBOSHOT yields better outcomes. In addition, we observe that the harmful coefficients decrease when the insight embeddings have less noise. In fact, we have that $\lim_{\sigma_{insight} \to 0} A_s = 0$ — the case of perfectly identifying harmful, helpful concepts.

Next, we provide a bound on $A_r$, the post-ROBOSHOT coefficient of a targeted helpful concept.

**Theorem 4.2.** *With an additional assumption $\alpha_s \leq 0$ $(1 \leq s \leq S)$ under the described noise model, the post-ROBOSHOT coefficient for helpful concept $r$ $(S + 1 \leq r \leq S + R)$ satisfies*

$$\mathbb{E} A_r \geq \left( 1 + \frac{\gamma_{r,r}^2}{\gamma_{r,r}^2 + (k-1)\sigma_{insight}^2} \right) \alpha_r.$$

Refer to Appendix C.3 for the proof. Theorem 4.2 implies the helpful coefficients are scaled up at a rate inversely proportional to the noise rate $\sigma_{insight}$. When concepts are perfectly identified, i.e. $\sigma_{insight} = 0$, the coefficient $\alpha_r$ is doubled, yielding more emphasis on the concept $z_r$ as desired.

## 5 EXPERIMENTAL RESULTS

This section evaluates the following claims:

- **Improving multimodal models (Section 5.1)**: ROBOSHOT improves zero-shot classification robustness of various multimodal models, even outperforming prompting techniques that include spurious insight descriptions (which we do not have access to) in the label prompts.

- **Improving language models (Section 5.2)**: ROBOSHOT improves zero-shot robustness using LM embeddings for text zero-shot classification, outperforming direct prompting to get predictions.

- **Label-free adaptation (Section 5.3)**: LFA (Algorithm 2) can further improve performance with only a small labeled set for validation (100 samples).

- **Extracting concepts from LM with varying capacities (Section 5.4)**: ROBOSHOT can extract insights from language models with varying capacities. Improvements persist with weaker LMs.

- **Ablations (Section 5.5)**: ROBOSHOT benefits from both removing harmful and boosting helpful representations (line 3 and line 7 in ROBOSHOT Algorithm 1).

**Metrics.** We use three metrics: average accuracy % (AVG), worst-group accuracy % (WG), and the gap between the two (Gap). While a model that relies on harmful correlations may achieve high AVG when such correlations are present in the majority of the test data, it may fail in settings where the correlation is absent. **A robust model should have high AVG and WG, with a small gap between them**.

**Baselines.** We compare against the following sets of baselines:

Table 1: Main results. Best WG and Gap performance **bolded**, second best underlined.

| Dataset | Model | ZS | | | GroupPrompt ZS | | | **ROBOSHOT** | | |
|---------|-------|-----|------|------|-----|------|------|-----|------|------|
| | | AVG | WG(↑) | Gap(↓) | AVG | WG(↑) | Gap(↓) | AVG | WG(↑) | Gap(↓) |
| Waterbirds | CLIP (ViT-B-32) | 80.7 | 27.9 | 52.8 | 81.6 | 43.5 | 38.1 | 82.0 | **54.4** | **28.6** |
| | CLIP (ViT-L-14) | 88.7 | 27.3 | 61.4 | 70.7 | 10.4 | 60.3 | 79.9 | **45.2** | **34.7** |
| | ALIGN | 72.0 | **50.3** | 21.7 | 72.5 | 5.8 | 66.7 | 50.9 | 41.0 | **9.9** |
| | AltCLIP | 90.1 | 35.8 | 54.3 | 82.4 | 29.4 | 53.0 | 78.5 | **54.8** | 23.7 |
| CelebA | CLIP (ViT-B-32) | 80.1 | 72.7 | 7.4 | 80.4 | 74.9 | 5.5 | 84.8 | **80.5** | **4.3** |
| | CLIP (ViT-L-14) | 80.6 | 74.3 | 6.3 | 77.9 | 68.9 | 9.0 | 85.5 | **82.6** | **2.9** |
| | ALIGN | 81.8 | 77.2 | 4.6 | 78.3 | 67.4 | 10.9 | 86.3 | **83.4** | **2.9** |
| | AltCLIP | 82.3 | **79.7** | **2.6** | 82.3 | 79.0 | 3.3 | 86.0 | 77.2 | 8.8 |
| PACS | CLIP (ViT-B-32) | 96.7 | 82.1 | 14.6 | 97.9 | 82.7 | 15.2 | 97.0 | **86.3** | **10.7** |
| | CLIP (ViT-L-14) | 98.1 | 79.8 | 18.3 | 98.2 | **86.6** | **11.6** | 98.1 | 83.9 | 14.2 |
| | ALIGN | 95.8 | **77.1** | **18.7** | 96.5 | 65.0 | 31.5 | 95.0 | 73.8 | 21.2 |
| | AltCLIP | 98.5 | 82.6 | 15.9 | 98.6 | 85.4 | 13.2 | 98.7 | **89.5** | **9.2** |
| VLCS | CLIP (ViT-B-32) | 75.6 | 20.5 | 55.1 | | - | | 76.5 | **33.0** | **43.5** |
| | CLIP (ViT-L-14) | 72.6 | 4.20 | 68.4 | | - | | 71.1 | **12.6** | **58.5** |
| | ALIGN | 78.8 | 33.0 | 45.8 | | - | | 77.6 | **39.8** | **37.8** |
| | AltCLIP | 78.3 | 24.7 | **53.6** | | - | | 78.9 | **25.0** | 53.9 |
| CXR14 | BiomedCLIP | 55.3 | 28.9 | 26.4 | | - | | 56.2 | **41.6** | **14.6** |

1. **Multimodal baselines**: (i) vanilla zero-shot classification (**ZS**) and (ii) ZS with group information (**Group Prompt ZS**). We use a variety of models: CLIP (ViT-B-32 and ViT-L-14) (Radford et al., 2021), ALIGN (Jia et al., 2021), and AltCLIP (Chen et al., 2022). Group Prompt ZS assumes access to spurious or harmful insight annotations and includes them in the label prompt. For instance, the label prompts for waterbirds dataset become [`waterbird with water background`, `waterbird with land background`, `landbird with water background`, `landbird with land background`]. We only report Group Prompt ZS results on datasets where spurious insight annotations are available.

2. **Language model baselines**: (i) zero-shot classification using language model embeddings, namely BERT (Reimers & Gurevych, 2019) and Ada (Neelakantan et al., 2022) (**ZS**), (ii) direct prompting to LMs, namely BART-MNLI (Lewis et al., 2019; Williams et al., 2018) and ChatGPT (Ziegler et al., 2019) (**Direct prompting**). We also compare with calibration methods for zero-shot text classification (Holtzman et al., 2021), results can be found in Appendix F.1.

## 5.1 IMPROVING MULTIMODAL MODELS

**Setup.** We experimented on five binary and multi-class datasets with spurious correlations and distribution shifts: **Waterbirds** (Sagawa et al., 2019), **CelebA** (Liu et al., 2015), **CXR14** (Wang et al., 2017), **PACS** (Li et al., 2017), and **VLCS** (Fang et al., 2013). Dataset details are provided in Appendix D.1. For CXR14, we use BiomedCLIP (Zhang et al., 2023)– a variant of CLIP finetuned on biomedical data. All experiments are conducted using frozen pretrained models embeddings. We evaluate on four model variants: **CLIP** (ViT-B-32 and ViT-L-14), **ALIGN**, and **AltCLIP**.

**Results.** Table 1 shows that **ROBOSHOT significantly improves the worst group performance (WG)** and maintains (and sometimes also improves) the overall average (AVG) without any auxiliary information (in contrast to Group Prompt, which requires access to spurious insight annotation). Improved robustness nearly across-the-board suggests that both the insights extracted from LMs and the representation modifications are useful. We also provide insights into the rare case where our method does not improve the baseline (e.g., ALIGN model on Waterbirds) in Appendix F.3.

## 5.2 IMPROVING LANGUAGE MODELS

**Setup.** We experimented on four text classification datasets: **CivilComments-WILDS** (Borkan et al., 2019; Koh et al., 2021), **HateXplain** (Mathew et al., 2021), **Amazon-WILDS** (Ni et al., 2019; Koh et al., 2021) and **Gender Bias** classification dataset (Dinan et al., 2020; Miller et al., 2017). We

Table 2: ROBOSHOT text zero-shot classification. Best WG **bolded**, second best underlined. We use inference models comparable to BERT embedding model (i.e., BART-MNLI) and to Ada embedding model (i.e., ChatGPT) for direct prompting experiments.

| Dataset | Model | ZS | | | Direct prompting | | | ROBOSHOT | | |
|---|---|---|---|---|---|---|---|---|---|---|
| | | AVG | WG($\uparrow$) | Gap($\downarrow$) | AVG | WG($\uparrow$) | Gap($\downarrow$) | AVG | WG($\uparrow$) | Gap($\downarrow$) |
| CivilComments | BERT | 48.1 | 33.3 | 14.8 | 32.5 | 15.7 | 16.8 | 49.7 | **42.3** | **7.4** |
| | Ada | 56.2 | 43.2 | 13.0 | 85.6 | 19.2 | 66.4 | 56.6 | **44.9** | **11.7** |
| HateXplain | BERT | 60.4 | 0.0 | 60.4 | 61.2 | 5.3 | 55.9 | 57.3 | **14.0** | **43.3** |
| | Ada | 62.8 | 14.3 | 48.5 | 55.4 | 12.2 | 43.2 | 63.6 | **21.1** | **42.5** |
| Amazon | BERT | 81.1 | 64.2 | 16.8 | 74.9 | 36.0 | 38.9 | 81.0 | **64.4** | **16.6** |
| | Ada | 81.2 | 63.4 | 17.8 | 80.1 | **73.5** | **6.6** | 82.9 | 63.8 | 19.1 |
| Gender Bias | BERT | 84.8 | 83.7 | 1.1 | 86.1 | 78.4 | 7.6 | 85.1 | **84.9** | **0.2** |
| | Ada | 77.9 | 60.0 | 17.9 | 90.1 | **86.6** | **3.5** | 78.0 | 60.1 | 17.9 |

use the default test splits of all datasets. In text experiments, the distinctions between harmful and helpful insights are less clear than for images– so here we only use harmful vector rejection (line 3 in ROBOSHOT). CivilComments and HateXplain are toxic classification datasets with unwanted correlation between toxicity labels and mentions of demographics (e.g., male, female, mentions of religions). The datasets are annotated with demographic mentions of each text, and we directly use them to construct $v^j$. For Amazon and Gender Bias datasets, we query LMs with task descriptions. All experiments are conducted using frozen pretrained model embedding. We provide full list of prompts used in Direct Prompting experiments in Appendix D.3.

**Results.** Table 2 shows that **ROBOSHOT also improves zero-shot text classification**, as shown by our consistent boost over the baselines across all datasets on BERT embedding model and BART-MNLI direct prompting. In the Gender Bias and Amazon experiments, RoboShot lifts weaker/older model performance to a level comparable to modern LLMs (ChatGPT).

## 5.3 LABEL-FREE ADAPTATION (LFA)

Next, we evaluate our technique for maximizing robustness when users have access to labeled validation data (as before, we do not use any training data). **Setup.** We run LFA (Algorithm 2) across

Table 3: LFA on CLIP ViT-B-32 embedding. Best WG **bolded**, second best underlined. Best WG in blue, best AVG in green

| Dataset | ROBOSHOT | | LFA | | LFA (100 val) | |
|---|---|---|---|---|---|---|
| | AVG | WG | AVG | WG | AVG | WG |
| Waterbirds | 82.0 | 54.5 | 83.8 $\pm$ 0.74 | **55.2** $\pm$ 0.75 | 84.2 $\pm$ 1.1 | 53.6 $\pm$ 1.76 |
| CelebA | 84.8 | 80.5 | 86.7 $\pm$ 0.811 | 83.4 $\pm$ 1.02 | 86.5 $\pm$ 0.72 | **83.8** $\pm$ 1.17 |
| PACS | 95.6 | 79.7 | 96.6 $\pm$ 0.43 | **84.3** $\pm$ 1.3 | 96.9 $\pm$ 0.38 | 82.5 $\pm$ 2.16 |
| VLCS | 74.1 | 25.0 | 76.3 $\pm$ 1.27 | 36.5 $\pm$ 5.0 | 77.0 $\pm$ 0.35 | **37.4** $\pm$ 3.34 |

5 different random seeds and report the mean and standard deviation test results from the model with the best validation performance. Table 3 shows results from using only 100 random validation samples (LFA 100 val) and the full validation set (LFA). We use WILDS (Koh et al., 2021) default splits in Waterbirds and CelebA, and randomly shuffle 70:20:10 train:test:validation splits in PACS and VLCS. Note that ROBOSHOT performance is slightly different from Table 1, because there we use all samples for test. The training was conducted using two NVIDIA RTX A4000 GPUs, and we report the hyperparameter choices in Appendix D.5.

**Results.** LFA gives extra improvements on both AVG and WG. Improvement mostly persists even when using only 100 validation samples. If users have more validation labels, performance can be

Table 4: ROBOSHOT with LMs of varying capacity. Best WG **bolded**, second best underlined

| Dataset | ZS | | Ours (ChatGPT) | | Ours (Flan-T5) | | Ours (GPT2) | | Ours (LLaMA) | |
|---|---|---|---|---|---|---|---|---|---|---|
| | AVG | WG | AVG | WG | AVG | WG | AVG | WG | AVG | WG |
| Waterbirds | 80.7 | 27.9 | 82.0 | **54.4** | 72.1 | 32.4 | 88.0 | 39.9 | 84.8 | 36.5 |
| CelebA | 80.1 | 72.7 | 84.8 | 80.5 | 77.5 | 68.2 | 80.3 | 74.1 | 84.2 | **82.0** |
| PACS | 96.7 | 82.1 | 97.0 | **86.3** | 96.2 | 80.3 | 97.2 | 74.0 | 94.8 | 71.9 |
| VLCS | 75.6 | 20.5 | 76.5 | **33.0** | 69.6 | 20.5 | 75.5 | 26.1 | 72.0 | 18.2 |

further improved. This indicates that *LFA can serve as a lightweight training-based alternative* to the fully zero-shot approach ROBOSHOT when a small set of labeled validation data is available.

## 5.4 EXTRACTING CONCEPTS FROM LMs WITH VARYING CAPACITIES

**Setup.** We use LMs with different capacities: **ChatGPT** (Ouyang et al., 2022), **Flan-T5** (Chung et al., 2022), **GPT2** (Radford et al., 2019), and **LLaMA** (Touvron et al., 2023), to obtain insights.

**Results.** Table 4 shows that even though the LM strength/sizes correlate with the performance, ROBOSHOT with weaker LMs still outperforms zero-shot baselines. We hypothesize, based on Theorem 4.1 and 4.2, that insight outputs from weaker/smaller LMs are still precise in specifying the useful and non-useful terms and thus ROBOSHOT is able to use the insight embeddings.

Table 5: Ablation. Best WG and Gap performance **bolded**, second best underlined.

| Dataset | Model | ZS | | | Ours ($v^j$ only) | | | Ours ($u^k$ only) | | | Ours (both) | | |
|---|---|---|---|---|---|---|---|---|---|---|---|---|---|
| | | AVG | WG(↑) | Gap(↓) | AVG | WG(↑) | Gap(↓) | AVG | WG(↑) | Gap(↓) | AVG | WG(↑) | Gap(↓) |
| Waterbirds | CLIP (ViT-B-32) | 80.7 | 27.9 | 52.8 | 82.0 | 50.4 | 31.6 | 82.6 | 30.2 | 52.4 | 83.0 | **54.4** | **28.6** |
| | CLIP (ViT-L-14) | 88.7 | 27.3 | 61.4 | 82.7 | 35.8 | 46.9 | 88.3 | 29.8 | 58.5 | 79.9 | **45.2** | **34.**7 |
| CelebA | CLIP (ViT-B-32) | 80.1 | 72.7 | 7.4 | 85.2 | **81.5** | **3.7** | 79.6 | 71.3 | 8.3 | 84.8 | 80.5 | 4.3 |
| | CLIP (ViT-L-14) | 80.6 | 74.3 | 6.3 | 85.9 | **82.8** | 3.1 | 80.0 | 73.1 | 6.9 | 85.5 | 82.6 | **2.9** |
| PACS | CLIP (ViT-B-32) | 96.7 | 82.1 | 14.6 | 97.0 | 83.7 | 13.3 | 96.6 | 84.2 | 12.4 | 97.0 | **86.3** | **10.7** |
| | CLIP (ViT-L-14) | 98.1 | 79.8 | 18.3 | 98.0 | 79.8 | 18.2 | 98.1 | 83.8 | 14.3 | 98.1 | **83.9** | **14.2** |
| VLCS | CLIP (ViT-B-32) | 75.6 | 20.5 | 55.1 | 75.6 | 22.7 | 52.9 | 76.4 | 29.5 | 46.9 | 76.5 | **33.0** | **43.5** |
| | CLIP (ViT-L-14) | 72.6 | 4.2 | 68.4 | 70.9 | 6.8 | 64.1 | 73.4 | 8.9 | 64.5 | 71.1 | **12.6** | **58.5** |
| CXR14 | BiomedCLIP | 55.3 | 28.9 | 26.4 | 55.7 | **41.8** | **13.9** | 54.8 | 21.8 | 33.0 | 56.2 | 41.6 | 14.6 |

## 5.5 ABLATIONS

**Setup.** We run ROBOSHOT with only harmful component mitigation (reject $v^j$: ROBOSHOT line 3), only boosting helpful vectors (amplify $u^k$: ROBOSHOT line 7), and both. Due to space constraint, we only include CLIP-based models ablations. Results on all models can be found in Appendix F.

**Results.** The combination of both projections often achieves the best performance, as shown in Table 5. Figure 2 provides insights into the impact of each projection. Rejecting $v^j$ reduces variance in one direction, while increasing $u^k$ amplifies variance in the orthogonal direction. When both projections are applied, they create a balanced mixture.

## 6 CONCLUSION

We introduced ROBOSHOT, a fine-tuning-free system that robustifies zero-shot pretrained models in a truly zero-shot way. Theoretically, we characterized the quantities required to obtain improvements over vanilla zero-shot classification. Empirically, we found that ROBOSHOT improves both multi-modal and language model zero-shot performance, has sufficient versatility to apply to various base models, and can use insights from less powerful language models.

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

## APPENDIX

The appendix contains additional related work, details, proofs, and experimental results. The glossary contains a convenient reminder of our terminology (Appendix A). Appendix C provides the proofs of theorems that appeared in Section 4. In Appendix D, we give more details and analysis of the experiments and provide additional experiment results. Finally, Appendix F entails additional experiments combining ROBOSHOT with other methods to highlight its versatility.

## A  GLOSSARY

The glossary is given in Table 6.

| Symbol | Definition |
|---|---|
| $x$ | input vector |
| $X$ | embedding matrix |
| $X_{proj}$ | ROBOSHOT projected embedding matrix |
| $y, \hat{y}$ | class label, prediction |
| $c^i$ | embedding of class $i$ |
| $z_1, \ldots, z_k$ | The concept vectors consisting of orthonormal vectors |
| $v^i, u^j$ | insight representations |
| $\alpha_j$ | The coefficient of input $x$ with respect to the concept $z_j$ (before ROBOSHOT) |
| $A_j$ | The coefficient of transformed input $\hat{x}$ with respect to the concept $z_j$ (after ROBOSHOT) |
| $\beta_{i,j}$ | The coefficient of $j$-th class embedding with respect to the concept $z_i$ |
| $\gamma_{i,j}$ | The coefficient of $j$-th insight vector with respect to the concept $z_i$ |
| $S$ | the number of harmful concepts |
| $R$ | the number of helpful concepts |
| $B$ | the number of benign concepts |
| $g$ | text encoder to get embeddings |
| $s^i$ | text string for insight vectors |
| $\sigma_{\text{benign}}, \sigma_{\text{insight}}$ | noise rates in the coefficients of benign/insight concepts |

Table 6: Glossary of variables and symbols used in this paper.

## B  EXTENDED RELATED WORK

**Using language to improve visual tasks.** A large body of work has shown the efficacy of using language to improve performance on vision tasks Radford et al. (2021); Frome et al. (2013); Le Cacheux et al. (2020). Most relevant are those that focus on robustness, such as Yang et al. (2023), which uses text descriptions of spurious attributes in a fine-tuning loss to improve robustness. In contrast to these works, we focus on using textual concepts to improve zero-shot model robustness—without fine-tuning. Other zero-shot works attempt to provide improvements to accuracy. For example, (Novack et al., 2023) increases zero-shot accuracy by first expanding the class options into their subclasses (e.g., dog into labrador and golden retriever) and producing a final prediction by mapping them back to the superclass. The most closely related of these to our work is Menon & Vondrick (2022); Maniparambil et al. (2023), where GPT-3 generated class descriptors are first generated, then CLIP predictions scores are grounded by additive decomposition of scores from the prompts with the descriptors. This approach also does not require fine-tuning. However, it focuses mainly on grounding through prompting with class descriptors, while ours focuses on removing harmful concepts and increasing helpful concepts in the embedding space—enabling improved robustness on difficult slices.

**Language models as priors.** The basis of our work is the observation that language models contain information that can serve as a prior for other tasks. Kıcıman et al. (2023) finds that LLMs can perform causal reasoning tasks, substantially outperforming existing methods. Choi et al. (2022) prompts LLMs for task-specific priors, leading to substantial performance improvements in feature selection, reinforcement learning, and causal discovery. Our work shares the spirit of these approaches in using the insights embedded in language models to enhance zero-shot robustness.

## C  THEORY DETAILS

### C.1  HARMFUL CONCEPT REMOVAL

As the simplest form of ROBOSHOT, we consider the case of ROBOSHOT the harmful concept removal only, without boosting helpful concepts. Recall our noise model:

$$x = \sum_{s=1}^{S} \alpha_s z_s + \sum_{r=S+1}^{S+R} \alpha_r z_r + \sum_{b=S+R+1}^{S+R+B} \alpha_b z_b$$

$$v^t = \sum_{s=1}^{S} \gamma_{s,t} z_s + \sum_{r=S+1}^{S+R} \gamma_{r,t} z_r + \sum_{b=S+R+1}^{S+R+B} \gamma_{b,t} z_b \qquad (1 \leq t \leq S).$$

Again, we assume that benign coefficients are drawn from a zero-centered Gaussian distribution, i.e. $\alpha_b, \gamma_{b,t} \sim \mathcal{N}(0, \sigma_{benign})$ and also helpful coefficients and non-target harmful coefficients are assumed to be drawn from a Gaussian distribution, i.e. $\gamma_{q,t} \sim \mathcal{N}(0, \sigma_{insight})$, where $1 \leq q \leq R$, $q \neq t$ so that only $\gamma_{t,t}$ is a constant.

### C.1.1  EFFECTS ON HARMFUL COEFFICIENTS

Now we prove the following theorem.

**Theorem C.1.** *Under the noise model described above, the post-removal coefficient $A_s$ for harmful concept $z_s$ satisfies*

$$|\mathbb{E}A_s| \leq \left| \frac{(k-1)\alpha_s \sigma_{insight}^2}{\gamma_{s,s}^2} \right| + \left| \sum_{t \neq s}^{S} \frac{\alpha_s \sigma_{insight}^2}{\gamma_{t,t}^2} \right|,$$

*where $k$ is the number of concepts ($k = S + R + B$).*

*Proof.* Let $\hat{x}$ be the output of harmful concept removal procedure such that

$$\hat{x} = x - \sum_{s=1}^{S} \frac{x^T v^s}{||v^s||^2} v^s$$

$$= \sum_{i=1}^{k} \alpha_i z_i - \sum_{s=1}^{S} \frac{\sum_{i}^{k} \alpha_i \gamma_{i,s}}{\sum_{l=1}^{k} \gamma_{l,s}^2} (\sum_{j=1}^{k} \gamma_{j,s} z_j)$$

As the first step, we sort out the coefficients of features. For notational convenience, let $T_s = \sum_{l=1}^{k} \gamma_{l,s}^2$. Then,

$$\hat{x} = \sum_{i=1}^{k} \alpha_i z_i - \sum_{s=1}^{S} \frac{\sum_{i=1}^{k} \alpha_i \gamma_{i,s}}{T_s} (\sum_{j=1}^{k} \gamma_{j,s} z_j)$$

$$= \sum_{i=1}^{k} \alpha_i z_i - \sum_{s=1}^{S} \sum_{i=1}^{k} \sum_{j=1}^{k} \frac{\alpha_i \gamma_{i,s} \gamma_{j,s}}{T_s} z_j$$

$$= \sum_{j=1}^{k} \alpha_j z_j - \sum_{j=1}^{k} \sum_{s=1}^{S} \sum_{i=1}^{k} \frac{\alpha_i \gamma_{i,s} \gamma_{j,s}}{T_s} z_j$$

$$= \sum_{j=1}^{k} \left( \alpha_j - \sum_{s=1}^{S} \sum_{i=1}^{k} \frac{\alpha_i \gamma_{i,s} \gamma_{j,s}}{T_s} \right) z_j$$

Thus we can get the expression for the coefficient of the target feature $z_s$ $(1 \le s \le S)$,

$$A_s = \alpha_s - \sum_{t=1}^{S} \sum_{i=1}^{k} \frac{\alpha_i \gamma_{i,t} \gamma_{s,t}}{T_t}$$

Next, we get the bound of the absolute expectation $|\mathbb{E} A_s|$.

$$|\mathbb{E} A_s| = \left| \mathbb{E} \alpha_s - \sum_{t=1}^{S} \sum_{i=1}^{k} \frac{\alpha_i \gamma_{i,t} \gamma_{s,t}}{\sum_{l=1}^{k} \gamma_{l,t}^2} \right|$$

$$\le \left| \mathbb{E} \alpha_s - \sum_{t=1}^{S} \frac{\alpha_s \gamma_{s,t}^2}{\sum_{l=1}^{k} \gamma_{l,t}^2} \right| + \left| \sum_{t=1}^{S} \mathbb{E} \frac{\sum_{i=1,i \ne s}^{S} \alpha_i \gamma_{i,t} \gamma_{s,t}}{\sum_{l=1}^{k} \gamma_{l,t}^2} \right|$$

Here, the second term on RHS is 0 by independence, i.e.

$$\left| \mathbb{E} \frac{\sum_{i=1,i \ne s}^{S} \alpha_i \gamma_{i,t} \gamma_{s,t}}{\sum_{l=1}^{k} \gamma_{l,t}^2} \right| \le \left| \mathbb{E} \frac{\sum_{i=1,i \ne s}^{k} \alpha_i \gamma_{i,t} \gamma_{s,t}}{\gamma_{t,t}^2} \right|$$

$$= \left| \sum_{i=1,i \ne s}^{k} \frac{\alpha_i}{\gamma_{t,t}^2} \mathbb{E} \gamma_{i,t} \gamma_{s,t} \right| = 0$$

since $\mathbb{E} \gamma_{s,t} \gamma_{j,t} = 0$ by independence. Now we split the first term and get the bounds separately.

$$|\mathbb{E}A_s| \le \left| \mathbb{E}\alpha_s - \sum_{t=1}^{S} \frac{\alpha_s \gamma_{s,t}^2}{\sum_{l=1}^{k} \gamma_{l,t}^2} \right|$$

$$\le \left| \mathbb{E}\alpha_s - \frac{\alpha_s \gamma_{s,s}^2}{\sum_{l=1}^{k} \gamma_{l,s}^2} \right| + \left| \sum_{t=1,t\ne s}^{S} \mathbb{E}\frac{\alpha_s \gamma_{s,t}^2}{\sum_{l=1}^{k} \gamma_{l,t}^2} \right|$$

The upper bound for the first term can be obtained by

$$\left| \mathbb{E}\alpha_s - \frac{\alpha_s \gamma_{s,s}^2}{\sum_{l=1}^{k} \gamma_{l,s}^2} \right| = \left| \mathbb{E} - \frac{\sum_{i\ne s}^{k} \alpha_s \gamma_{i,s}^2}{\sum_{l=1}^{k} \gamma_{l,s}^2} \right|$$

$$\le \left| \mathbb{E} \frac{\sum_{i\ne s}^{k} \alpha_s \gamma_{i,s}^2}{\gamma_{s,s}^2} \right|$$

$$\le \left| \frac{\alpha_s}{\gamma_{s,s}^2} \sum_{i\ne s}^{k} \mathbb{E}\gamma_{i,s}^2 \right|$$

$$\le \left| \frac{(k-1)\alpha_s \sigma_{insight}^2}{\gamma_{s,s}^2} \right|.$$

And, for the second term,

$$\left| \sum_{t=1,t\ne s}^{S} \mathbb{E}\frac{\alpha_s \gamma_{s,t}^2}{\sum_{i=1}^{k} \gamma_{i,t}^2} \right| \le \left| \sum_{t=1,t\ne s}^{S} \mathbb{E}\frac{\alpha_s \gamma_{s,t}^2}{\gamma_{t,t}^2} \right|$$

$$= \left| \sum_{t=1,t\ne s}^{S} \frac{\alpha_s}{\gamma_{t,t}^2} \mathbb{E}\gamma_{s,t}^2 \right|$$

$$= \left| \sum_{t\ne s}^{S} \frac{\alpha_s \sigma_{insight}^2}{\gamma_{t,t}^2} \right|$$

Combining two bounds, we get the proposed result.

$$|\mathbb{E}A_s| \le \left| \frac{(k-1)\alpha_s \sigma_{insight}^2}{\gamma_{s,s}^2} \right| + \left| \sum_{t\ne s}^{S} \frac{\alpha_s \sigma_{insight}^2}{\gamma_{t,t}^2} \right|.$$

$\square$

While the constant $(k-1)$ can look daunting since it actually increases as the number of concepts increases, a bound less affected by $\sigma_{insight}^2$ exists as well, scaling down the target coefficient $\alpha_s$.

**Corollary C.1.1.** *Under the noise model of Theorem C.1, the post-removal coefficient for harmful concept $s$ satisfies*

$$|\mathbb{E}A_s| \le \left| \alpha_s \frac{(k-1)\sigma_{insight}^2}{\gamma_{s,s}^2 + (k-1)\sigma_{insight}^2} \right| + \left| \sum_{t\ne s}^{S} \frac{\alpha_s \sigma_{insight}^2}{\gamma_{t,t}^2} \right|,$$

*where $k$ is the number of concepts ($k = S + R + B$).*

*Proof.* With the identical steps to the proof of Theorem C.1, we can obtain

$$
|\mathbb{E}A_s| \leq \left| \mathbb{E}\alpha_s - \sum_{t=1}^{S} \frac{\alpha_s \gamma_{s,t}^2}{\sum_{l=1}^{k} \gamma_{l,t}^2} \right|
$$

$$
\leq \left| \mathbb{E}\alpha_s - \frac{\alpha_s \gamma_{s,s}^2}{\sum_{l=1}^{k} \gamma_{l,s}^2} \right| + \left| \sum_{t=1,t\neq s}^{S} \mathbb{E} \frac{\alpha_s \gamma_{s,t}^2}{\sum_{l=1}^{k} \gamma_{l,t}^2} \right|
$$

$$
\leq \left| \mathbb{E}\alpha_s - \frac{\alpha_s \gamma_{s,s}^2}{\sum_{l=1}^{k} \gamma_{l,s}^2} \right| + \left| \sum_{t=1,t\neq s}^{S} \frac{\alpha_s}{\gamma_{t,t}^2} \mathbb{E}\gamma_{s,t}^2 \right|.
$$

We improve the first term as follows.

$$
\left| \mathbb{E}\alpha_s - \frac{\alpha_s \gamma_{s,s}^2}{\sum_{l=1}^{k} \gamma_{l,s}^2} \right| = \left| \alpha_s - \alpha_s \gamma_{s,s}^2 \mathbb{E} \frac{1}{\sum_{l=1}^{k} \gamma_{l,s}^2} \right|
$$

$$
\leq \left| \alpha_s - \alpha_s \gamma_{s,s}^2 \frac{1}{\mathbb{E}\sum_{l=1}^{k} \gamma_{l,s}^2} \right| \quad \because \text{Jensen's inequality } \mathbb{E}\frac{1}{\sum_{l=1}^{k} \gamma_{l,s}^2} \geq \frac{1}{\mathbb{E}\sum_{l=1}^{k} \gamma_{l,s}^2}
$$

$$
= \left| \alpha_s \left( 1 - \frac{\gamma_{s,s}^2}{\mathbb{E}\sum_{l=1}^{k} \gamma_{l,s}^2} \right) \right|
$$

$$
= \left| \alpha_s \left( 1 - \frac{\gamma_{s,s}^2}{\gamma_{s,s}^2 + (k-1)\sigma_{insight}^2} \right) \right|
$$

$$
= \left| \alpha_s \left( \frac{(k-1)\sigma_{insight}^2}{\gamma_{s,s}^2 + (k-1)\sigma_{insight}^2} \right) \right|.
$$

$\square$

### C.1.2 EFFECTS ON HELPFUL, BENIGN COEFFICIENTS

Based on the coefficient expression

$$
A_q = \alpha_q - \sum_{t=1}^{S} \sum_{i=1}^{k} \frac{\alpha_i \gamma_{i,t} \gamma_{q,t}}{\sum_{l=1}^{k} \gamma_{l,t}^2},
$$

we analyze the bound of $|\mathbb{E}A_q|$ for $S+1 \leq q \leq k$. Essentially, the following theorem implies helpful, benign coefficients are less affected than harmful coefficients as long as the harmful coefficients of insight embeddings are significant and the noise is small.

**Theorem C.2.** *Under the same noise model described above, the post-removal coefficient for helpful or benign concept $q$ satisfies*

$$
|\mathbb{E}A_q - \alpha_q| \leq \left| \sum_{t=1}^{S} \frac{\alpha_q \sigma_{insight}^2}{\gamma_{t,t}^2} \right|.
$$

*Proof.* The proof technique is essentially identical to Theorem C.1.

$$
\begin{aligned}
|\mathbb{E}A_q - \alpha_q| &= \left| \alpha_q - \mathbb{E}\alpha_q - \sum_{t=1}^{S} \frac{\alpha_q \gamma_{q,t}^2 + \sum_{j=1, j\neq q} \alpha_q \gamma_{q,t}\gamma_{j,t}}{\sum_{l=1}^{k} \gamma_{l,t}^2} \right| \\
&\leq \left| \mathbb{E}\sum_{t=1}^{S} \frac{\alpha_q \gamma_{q,t}^2}{\sum_{l=1}^{k} \gamma_{l,t}^2} \right| + \left| \mathbb{E} \frac{\sum_{j=1, j\neq q} \alpha_q \gamma_{q,t}\gamma_{j,t}}{\sum_{l=1}^{k} \gamma_{l,t}^2} \right| \\
&= \left| \mathbb{E}\sum_{t=1}^{S} \frac{\alpha_q \gamma_{q,t}^2}{\sum_{l=1}^{k} \gamma_{l,t}^2} \right| \quad \because \left| \mathbb{E} \frac{\sum_{j=1, j\neq q} \alpha_q \gamma_{q,t}\gamma_{j,t}}{\sum_{l=1}^{k} \gamma_{l,t}^2} \right| = 0 \\
&\leq \left| \sum_{t=1}^{S} \frac{\alpha_q}{\gamma_{t,t}^2} \mathbb{E}\gamma_{q,t}^2 \right| \\
&= \left| \sum_{t=1}^{S} \frac{\alpha_q \sigma_{insight}^2}{\gamma_{t,t}^2} \right|.
\end{aligned}
$$

$\square$

This bound implies the differences of helpful or benign features by harmful concept removal are proportional to the noise of insight embeddings $\sigma_{insight}^2$, and inversely proportional to the coefficients of harmful coefficients of insight embeddings.

## C.2 HELPFUL CONCEPT ADDITION

With a similar fashion to the harmful concept removal, we consider the following noise model for the helpful concept addition.

$$
x = \sum_{s=1}^{S} \alpha_s z_s + \sum_{r=S+1}^{S+R} \alpha_r z_r + \sum_{b=S+R+1}^{S+R+B} \alpha_b z_b
$$

$$
v^t = \sum_{s=1}^{S} \gamma_{s,t} z_s + \sum_{r=S+1}^{S+R} \gamma_{r,t} z_r + \sum_{b=S+R+1}^{S+R+B} \gamma_{b,t} z_b \qquad (S+1 \leq t \leq S+R)
$$

. Again, we assume that benign coefficients are drawn from a zero-centered Gaussian distribution, i.e. $\alpha_b, \gamma_{b,t} \sim \mathcal{N}(0, \sigma_{benign})$ and also harmful coefficients and non-target helpful coefficients are assumed to be drawn from another Gaussian distribution, i.e. $\gamma_{q,t} \sim \mathcal{N}(0, \sigma_{insight})$, where $1 \leq q \leq S+R, q \neq t$ so that only $\gamma_{t,t}$ are constants.

### C.2.1 LOWER BOUND FOR THE COEFFICIENT OF HELPFUL CONCEPT

**Theorem C.3.** *Under the described noise model, the post-addition coefficient for helpful concept $r$ satisfies*

$$
\mathbb{E}A_r \geq \left( 1 + \frac{\gamma_{r,r}^2}{\gamma_{r,r}^2 + (k-1)\sigma_{insight}^2} \right) \alpha_r.
$$

*Proof.* Let $\hat{x}$ be the output of helpful concept addition procedure such that

$$
\begin{aligned}
\hat{x} &= x + \sum_{t=S+1}^{S+R} \frac{x^T v^t}{\|v^t\|^2} v^t \\
&= \sum_{i=1}^{k} \alpha_i z_i + \sum_{t=S+1}^{S+R} \frac{\sum_{i=1}^{k} \alpha_i \gamma_{i,t}}{\sum_{l=1}^{k} \gamma_{l,t}^2} \left( \sum_{j=1}^{k} \gamma_{j,t} z_j \right).
\end{aligned}
$$

As the first step, we sort out the coefficients of concepts. For notational convenience, let $T_t = \sum_{l=1}^{k} \gamma_{l,t}^2$. Then,

$$
\begin{aligned}
\hat{x} &= \sum_{i=1}^{k} \alpha_i z_i + \sum_{t=S+1}^{S+R} \frac{\sum_{i=1}^{k} \alpha_i \gamma_{i,t}}{T_t} \left( \sum_{j=1}^{k} \gamma_{j,t} z_j \right) \\
&= \sum_{i=1}^{k} \alpha_i z_i + \sum_{t=S+1}^{S+R} \sum_{i=1}^{k} \sum_{j=1}^{k} \frac{\alpha_i \gamma_{i,t} \gamma_{j,t}}{T_t} z_j \\
&= \sum_{j=1}^{k} \alpha_j z_j + \sum_{j=1}^{k} \sum_{t=S+1}^{S+R} \sum_{i=1}^{k} \frac{\alpha_i \gamma_{i,t} \gamma_{j,t}}{T_t} z_j \\
&= \sum_{j=1}^{k} \left( \alpha_j + \sum_{t=S+1}^{S+R} \sum_{i=1}^{k} \frac{\alpha_i \gamma_{i,t} \gamma_{j,t}}{T_t} \right) z_j.
\end{aligned}
$$

Thus we can get the expression for the coefficient of the target concept $z_r \quad (S+1 \le r \le S+R)$,

$$
A_r = \alpha_r + \sum_{t=S+1}^{S+R} \sum_{i=1}^{k} \frac{\alpha_i \gamma_{i,t} \gamma_{r,t}}{T_t}.
$$

Then,

$$
\begin{aligned}
\mathbb{E} A_r &= \mathbb{E} \alpha_r + \sum_{t=S+1}^{S+R} \sum_{i=1}^{k} \frac{\alpha_i \gamma_{i,t} \gamma_{r,t}}{T_t} \\
&= \alpha_r + \sum_{t=S+1}^{S+R} \sum_{i=1}^{k} \mathbb{E} \frac{\alpha_i \gamma_{i,t} \gamma_{r,t}}{\sum_{l=1}^{k} \gamma_{l,t}^2} \\
&= \alpha_r + \mathbb{E} \frac{\alpha_r \gamma_{r,r}^2}{\sum_{l=1}^{k} \gamma_{l,r}^2} + \sum_{i=1,i\neq r}^{k} \mathbb{E} \frac{\alpha_i \gamma_{i,r} \gamma_{r,r}}{\sum_{l=1}^{k} \gamma_{l,r}^2} + \sum_{t=S+1,t\neq r}^{S+R} \sum_{i=1}^{k} \mathbb{E} \frac{\alpha_i \gamma_{i,t} \gamma_{r,t}}{\sum_{l=1}^{k} \gamma_{l,t}^2} \\
&= \alpha_r + \mathbb{E} \frac{\alpha_r \gamma_{r,r}^2}{\sum_{l=1}^{k} \gamma_{l,r}^2} + \sum_{i=1,i\neq r}^{k} \gamma_{r,r} \mathbb{E} \frac{\alpha_i \gamma_{i,r}}{\sum_{l=1}^{k} \gamma_{l,r}^2} + \sum_{t=S+1,t\neq r}^{S+R} \sum_{i=1}^{k} \mathbb{E} \frac{\alpha_i \gamma_{i,t} \gamma_{r,t}}{\sum_{l=1}^{k} \gamma_{l,t}^2} \\
&= \alpha_r + \mathbb{E} \frac{\alpha_r \gamma_{r,r}^2}{\sum_{l=1}^{k} \gamma_{l,r}^2} + \sum_{t=S+1,t\neq r}^{S+R} \sum_{i=1}^{k} \mathbb{E} \frac{\alpha_i \gamma_{i,t} \gamma_{r,t}}{\sum_{l=1}^{k} \gamma_{l,t}^2} \quad \because \text{by symmetry} \\
&= \alpha_r + \mathbb{E} \frac{\alpha_r \gamma_{r,r}^2}{\sum_{l=1}^{k} \gamma_{l,r}^2} \quad \because \text{by law of total expectation and symmetry} \\
&\ge \alpha_r + \alpha_r \gamma_{r,r}^2 \mathbb{E} \frac{1}{\sum_{l=1}^{k} \gamma_{l,r}^2} \\
&\ge \alpha_r + \alpha_r \gamma_{r,r}^2 \frac{1}{\mathbb{E} \sum_{l=1}^{k} \gamma_{l,r}^2} \quad \because \text{Jensen's inequality} \\
&= \alpha_r + \alpha_r \gamma_{r,r}^2 \frac{1}{\gamma_{r,r}^2 + (k-1)\sigma_{insight}^2}.
\end{aligned}
$$

Thus, we obtain the result.

$$
\mathbb{E} A_r \ge \left( 1 + \frac{\gamma_{r,r}^2}{\gamma_{r,r}^2 + (k-1)\sigma_{insight}^2} \right) \alpha_r.
$$

$\square$

### C.2.2  EFFECTS ON HARMFUL, BENIGN COEFFICIENTS

For notational convenience, let $I_{helpful}^c$ be the non-helpful concept index set such that $I_{helpful}^c = \{i \in \mathbb{N} | i \leq S \text{ or } S + R + 1 \leq i \leq S + R + B\}$. For $q \in I_R^c$, we obtain the bound of effects on harmful, benign coefficients with a similar fashion to the harmful concept removal case.

**Theorem C.4.** *Under the same noise model described above, the post-addition coefficient for help-ful or benign concept q satisfies*

$$|\mathbb{E}A_q - \alpha_q| \leq \left| \sum_{t=S+1}^{S+R} \frac{\alpha_q \sigma_{insight}^2}{\gamma_{t,t}^2} \right|.$$

*Proof.*

$$|\mathbb{E}A_q - \alpha_q| = \left| \alpha_q - \mathbb{E}\alpha_q + \sum_{t=1}^{S} \frac{\alpha_q \gamma_{q,t}^2 + \sum_{j=1, j \neq q} \alpha_q \gamma_{q,t} \gamma_{j,t}}{\sum_{l=1}^{k} \gamma_{l,t}^2} \right|$$

$$\leq \left| \mathbb{E} \sum_{t=S+1}^{S+R} \frac{\alpha_q \gamma_{q,t}^2}{\sum_{l=1}^{k} \gamma_{l,t}^2} \right| + \left| \mathbb{E} \frac{\sum_{j=1, j \neq q} \alpha_q \gamma_{q,t} \gamma_{j,t}}{\sum_{l=1}^{k} \gamma_{l,t}^2} \right|$$

$$= \left| \mathbb{E} \sum_{t=S+1}^{S+R} \frac{\alpha_q \gamma_{q,t}^2}{\sum_{l=1}^{k} \gamma_{l,t}^2} \right| \quad \because \left| \mathbb{E} \frac{\sum_{j=1, j \neq q} \alpha_q \gamma_{q,t} \gamma_{j,t}}{\sum_{l=1}^{k} \gamma_{l,t}^2} \right| = 0$$

$$\leq \left| \sum_{t=S+1}^{S+R} \frac{\alpha_q}{\gamma_{t,t}^2} \mathbb{E}\gamma_{q,t}^2 \right|$$

$$= \left| \sum_{t=S+1}^{S+R} \frac{\alpha_q \sigma_{insight}^2}{\gamma_{t,t}^2} \right|.$$

$\square$

### C.3  COMBINED MAIN RESULTS

Now, we are ready to provide the combine main result, i.e. the coefficient bounds with harmful concept removal and helpful concept addition. The noise model can be described as follows.

$$x = \sum_{s=1}^{S} \alpha_s z_s + \sum_{r=S+1}^{S+R} \alpha_r z_r + \sum_{b=S+R+1}^{S+R+B} \alpha_b z_b$$

$$v^t = \sum_{s=1}^{S} \gamma_{s,t} z_s + \sum_{r=S+1}^{S+R} \gamma_{r,t} z_r + \sum_{b=S+R+1}^{S+R+B} \gamma_{b,t} z_b \qquad (1 \leq t \leq S + R)$$

$$\alpha_b, \gamma_{b,t} \sim \mathcal{N}(0, \sigma_{benign})$$

$$\gamma_{q,t} \sim \mathcal{N}(0, \sigma_{insight}),$$

where $1 \leq q \leq S + R$, $q \neq s$ so that only $\gamma_{t,t}$ is a constant. We can obtain the expression for each coefficient as before.

$$\hat{x} = \sum_{j=1} \left( a_j - \sum_{s=1}^{S} \sum_{i=1}^{k} \frac{\alpha_i \gamma_{i,s} \gamma_{j,s}}{T_s} + \sum_{r=S+1}^{S+R} \sum_{i=1}^{k} \frac{\alpha_i \gamma_{i,r} \gamma_{j,r}}{T_r} \right) z_j$$

$$A_q = a_q - \sum_{s=1}^{S} \sum_{i=1}^{k} \frac{\alpha_i \gamma_{i,s} \gamma_{q,s}}{T_s} + \sum_{r=S+1}^{S+R} \sum_{i=1}^{k} \frac{\alpha_i \gamma_{i,r} \gamma_{q,r}}{T_r},$$

where $A_q$ is the coefficient of $z_q (1 \leq q \leq k)$ after ROBOSHOT(ignoring normalization) and $T_t = \sum_{l=1}^{k} \gamma_{l,t}^2$. Using the results from the previous subsections, we provide an upper bound on harmful coefficients, a lower bound on helpful coefficients, and an upper bound on the change in the benign coefficients. We restate Theorem 4.1, 4.2 and provide proofs.

**Theorem 4.1.** *Under the combined noise model described above, the post-*ROBOSHOT *coefficient for harmful concept $q$ $(1 \leq q \leq S)$ satisfies*

$$|\mathbb{E}A_q| \leq \left| \frac{(k-1)\alpha_q \sigma_{insight}^2}{\gamma_{q,q}^2} \right| + \left| \sum_{t=1, t \neq q}^{S+R} \frac{\alpha_q \sigma_{insight}^2}{\gamma_{t,t}^2} \right|,$$

*where $k$ is the number of concepts ($k = S + R + B$).*

*Proof.*

$$
\begin{aligned}
|\mathbb{E}A_q| &= \left| \mathbb{E}a_q - \sum_{s=1}^{S} \sum_{i=1}^{k} \frac{\alpha_i \gamma_{i,s} \gamma_{q,s}}{T_s} + \sum_{r=S+1}^{S+R} \sum_{i=1}^{k} \frac{\alpha_i \gamma_{i,r} \gamma_{q,r}}{T_r} \right| \\
&\leq \left| \frac{(k-1)\alpha_q \sigma_{insight}^2}{\gamma_{q,q}^2} \right| + \left| \sum_{s=1, s \neq q}^{S} \frac{\alpha_q \sigma_{insight}^2}{\gamma_{s,s}^2} \right| + \left| \sum_{t=S+1}^{S+R} \frac{\alpha_q \sigma_{insight}^2}{\gamma_{t,t}^2} \right| \\
&= \left| \frac{(k-1)\alpha_q \sigma_{insight}^2}{\gamma_{q,q}^2} \right| + \left| \sum_{t=1, t \neq q}^{S+R} \frac{\alpha_q \sigma_{insight}^2}{\gamma_{t,t}^2} \right| \quad \because \text{two terms have the same sign by } a_q
\end{aligned}
$$

$\square$

Next, we state the lower bound for the helpful features. We assume the signs of harmful concepts in input embeddings

$$\alpha_s \leq 0 \quad (1 \leq s \leq S),$$

to keep the appearance of the result clear.

**Theorem 4.2.** *With an additional assumptions $\alpha_s \leq 0$ $(1 \leq s \leq S)$ under the combined noise model, the post-*ROBOSHOT *coefficient for helpful concept $q(S + 1 \leq q \leq S + R)$ satisfies*

$$\mathbb{E}A_q \geq \left( 1 + \frac{\gamma_{q,q}^2}{\gamma_{q,q}^2 + (k-1)\sigma_{insight}^2} \right) \alpha_q.$$

*Proof.*

$$
\begin{aligned}
\mathbb{E}A_q &= \mathbb{E}a_q - \sum_{s=1}^{S} \sum_{i=1}^{k} \frac{\alpha_i \gamma_{i,s} \gamma_{q,s}}{T_s} + \sum_{r=S+1}^{S+R} \sum_{i=1}^{k} \frac{\alpha_i \gamma_{i,r} \gamma_{q,r}}{T_r} \\
&= \mathbb{E}a_q + \sum_{r=S+1}^{S+R} \sum_{i=1}^{k} \frac{\alpha_i \gamma_{i,r} \gamma_{q,r}}{T_r} - \mathbb{E} \sum_{s=1}^{S} \sum_{i=1}^{k} \frac{\alpha_i \gamma_{i,s} \gamma_{q,s}}{T_s} \\
&= \mathbb{E}a_q + \sum_{r=S+1}^{S+R} \sum_{i=1}^{k} \frac{\alpha_i \gamma_{i,r} \gamma_{q,r}}{T_r} - \mathbb{E} \sum_{s=1}^{S} \frac{\alpha_s \gamma_{q,s}^2}{T_s} - \mathbb{E} \sum_{s=1}^{S} \sum_{i=1, i \neq q}^{k} \frac{\alpha_i \gamma_{i,s} \gamma_{q,s}}{T_s}.
\end{aligned}
$$

Here, $\mathbb{E}\sum_{s=1}^{S} \sum_{i=1, i \neq q}^{k} \frac{\alpha_i \gamma_{i,s} \gamma_{q,s}}{T_s} = 0$ by symmetry and law of total expectation, and $-\mathbb{E}\sum_{s=1}^{S} \frac{\alpha_s \gamma_{q,s}^2}{T_s} \geq 0$ since $\alpha_s \leq 0$ by assumption, which can be dropped for a lower bound.

$$\mathbb{E}A_q = \mathbb{E}a_q + \sum_{r=S+1}^{S+R}\sum_{i=1}^{k}\frac{\alpha_i\gamma_{i,r}\gamma_{q,r}}{T_r} - \mathbb{E}\sum_{s=1}^{S}\frac{\alpha_s\gamma_{q,s}^2}{T_s} - \mathbb{E}\sum_{s=1}^{S}\sum_{i=1,i\neq q}^{k}\frac{\alpha_i\gamma_{i,s}\gamma_{q,s}}{T_s}$$

$$\geq \mathbb{E}a_q + \sum_{r=S+1}^{S+R}\sum_{i=1}^{k}\frac{\alpha_i\gamma_{i,r}\gamma_{q,r}}{T_r}$$

$$\geq \left(1 + \frac{\gamma_{q,q}^2}{\gamma_{q,q}^2 + (k-1)\sigma_{insight}^2}\right)\alpha_q.$$

$\square$

Now, we state the upper bound on the changes in benign concepts. The proof is straightforward from the previous ones in harmful concept removal and helpful concept addition.

**Corollary C.4.1.** *Under the same combined noise model, the post-*RoboShot *coefficient for benign concept $q$ satisfies*

$$|\mathbb{E}A_q - \alpha_q| \leq \left|\sum_{t=1}^{S+R}\frac{\alpha_q\sigma_{insight}^2}{\gamma_{t,t}^2}\right|.$$

# D EXPERIMENTS DETAILS

## D.1 DATASETS

Table 7 provides details of the datasets used in our experiments. For Gender Bias dataset Dinan et al. (2020); Miller et al. (2017), we test using the train set to get more data. For all other datasets, we use the default test set. For Amazon-WILDS Ni et al. (2019) dataset, we convert the original 5-class rating classification into binary, by removing all samples with rating 3, and convert rating 1 and 2 into *bad* label, and 4 and 5 into *good* label.

| Dataset | Groups | $N_{all}$ | $N_{wg}$ | $n_{class}$ | classes |
|---|---|---|---|---|---|
| Waterbirds | { landbird in land, landbird in water, waterbird on land, waterbird on water } | 5794 | 642 | 2 | {landbird, waterbird } |
| CelebA | { male & not blond, female & not blond, male & blond , female & blond } | 19962 | 180 | 2 | {not blond, blond} |
| PACS | { art, cartoons, photos, sketches,} | 9991 | 80 | 7 | {dogs, elphant, giraffe, guitar, house, person } |
| VLCS | { Caltech101, LabelMe, SUN09, VOC2007 } | 10725 | 20 | 5 | {bird, car, chair, dog, person} |
| CXR14 | { no-pneumothorax, pneumothorax } | 2661 | 20 | 2 | {no-pneumothorax, pneumothorax} |
| CivilComments-WILDS | {male, female, LGBTQ, christian, muslim, other religions, black, white } | 133782 | 520 | 2 | {non-toxic, toxic } |
| HateXplain | {hindu, islam, minority, refugee, indian, caucasian, hispanic, women, disability, homosexual, arab, christian, jewish, men, african, nonreligious, asian, indigenous, heterosexual, buddhism, bisexual, asexual} | 1921 | 6 | 2 | {normal, offensive} |
| Amazon-WILDS | {beauty, garden, books, luxury beauty, kindle store, movies and TV, pet supplies, industrial and scientific, office products, CDs and vinyl, electronics, cell phones, magazine, clothing, groceries, music, instruments, tools, sports, automotive, toys, arts crafts, kitchen, video games, pantry, software, gift cards } | 90078 | 25 | 2 | {good,bad} |
| Gender Bias | {male, female } | 22750 | 3594 | 2 | {female, male} |

Table 7: Dataset details

| Dataset | Model | $v^{harmful}$ prompt | $v^{helpful}$ prompt |
|---|---|---|---|
| All | ChatGPT | "List the biased/spurious differences between [classes]." | "List the true visual differences between [classes]." |
| | Flan-T5 & GPT2 | {"[class] typically", "[class] usually"} | {"a characteristic of [class]: ", "[class] are", ""a [class] is", "Charactericstics of [class]" "Stereotype of [class]" "Typical characteristic of [class]"} |
| | LLaMA | "List the biased/spurious characteristics of [class]" | "List the visual characteristics of [class]" |

Table 8: Image dataset prompt details

| Dataset | Model | $v^{harmful}$ prompt |
|---|---|---|
| Amazon-WILDS | ChatGPT | "what are the biased differences between good and bad amazon reviews?" |
| Gender bias | ChatGPT | "what are the biased differences between comments about female and comments about male?" |

Table 9: NLP dataset prompt details

| Dataset | Class prompt |
|---|---|
| Waterbirds | [ "a landbird", "a waterbird" ] |
| CelebA | [ "person with dark hair", "person with blond hair" ] |
| PACS | "an image of [class]" |
| VLCS | "this object is [class]" |
| CXR14 | [ "non-pneumothorax", "pneumothorax" ] |
| CivilComments-WILDS | [ "non-toxic", "toxic" ] |
| HateXplain | [ "normal", "offensive" ] |
| Amazon-WILDS | [ "negative", "positive" ] |
| Gender Bias | [ "female", "male" ] |

Table 10: Class prompt details

## D.2 PROMPT TEMPLATES

We provide details on prompts used to get the $v^{harmful}$ and $v^{helpful}$ on image datasets in Table 8. As mentioned in the main body, for NLP datasets we only used $v^{harmful}$. Additionally, we use the demographic mentions annotations to construct $v^{harmful}$ in CivilComments-WILDS Borkan et al. (2019); Koh et al. (2021) and HateXplain Mathew et al. (2021). We provide prompt details to get $v^{harmful}$ for Amazon-WILDS Ni et al. (2019); Koh et al. (2021) and Gender Bias Dinan et al. (2020); Miller et al. (2017) datasets in Table 9. We also provide class prompts in Table 10.

## D.3 DIRECT PROMPTING PROMPT TEMPLATES

Table 11 shows the prompts used for Direct Prompting ChatGPT baseline in Table 2. For BART-MNLI, we directly use the dataset labels as label input to the model.

| Dataset | Prompt |
|---|---|
| CivilComments-WILDS | Is this a toxic comment? Answer with yes/no |
| HateXplain | Is this a toxic comment? Answer with yes/no |
| Amazon-WILDS | Is this a positive or negative product review? Answer with pos/neg |
| Gender Bias | Is this text about male/female? Answer with male/female |

Table 11: Direct prompting prompts for ChatGPT

## D.4 ROBOSHOT EXPERIMENT DETAILS

All ROBOSHOT experiments are carried out using frozen weights and embeddings from huggingface (ALIGN, AltCLIP) and open-clip (CLIP ViT-B-32 and ViT-L-14, BiomedCLIP), and no training is involved. There is no randomness in the ROBOSHOT experiment results reported in the main body of the paper.

## D.5 LFA EXPERIMENT DETAILS

| Dataset | Batch size | Learning rate |
|---|---|---|
| Waterbirds | $\{1.5e^{-8}, 2.5e^{-8}, 5e^{-8}, 2.5e^{-7}\}$ | $\{16, 32, 64\}$ |
| CelebA | $\{7.5e^{-9}, 1e^{-8}, 2.5e^{-8}\}$ | $\{16, 32, 64\}$ |
| PACS | $\{2.5e^{-9}, 5e^{-9}, 7.5e^{-9}, 1.5e^{-8}\}$ | $\{16, 32, 64\}$ |
| VLCS | $\{2.5e^{-9}, 5e^{-9}, 7.5e^{-9}, 1.5e^{-8}\}$ | $\{16, 32, 64\}$ |

Table 12: LFA hyperparameter choices

Table 12 shows the choices of hyperparameters we tune over for LFA experiments. We use SGD optimizer with fixed default momentum form PyTorch. All training are run for a fixed maximum epoch of 300, and we choose model based on validation performance.

## E    FULL ABLATION RESULT

Table 13: Ablation. Best WG and Gap performance **bolded**, second best underlined.

| Dataset | Model | ZS | | | Ours ($v^j$ only) | | | Ours ($u^k$ only) | | | Ours (both) | | |
|---|---|---|---|---|---|---|---|---|---|---|---|---|---|
| | | AVG | WG(↑) | Gap(↓) | AVG | WG(↑) | Gap(↓) | AVG | WG(↑) | Gap(↓) | AVG | WG(↑) | Gap(↓) |
| Waterbirds | CLIP (ViT-B-32) | 80.7 | 27.9 | 52.8 | 82.0 | 50.4 | 31.6 | 82.6 | 30.2 | 52.4 | 83.0 | **54.4** | **28.6** |
| | CLIP (ViT-L-14) | 88.7 | 27.3 | 61.4 | 82.7 | 35.8 | 46.9 | 88.3 | 29.8 | 58.5 | 79.9 | **45.2** | **34.7** |
| | ALIGN | 72.0 | 50.3 | 21.7 | 56.4 | 41.6 | 14.8 | 62.8 | **56.4** | **6.4** | 50.9 | 41.0 | 9.9 |
| | AltCLIP | 90.1 | 35.8 | 54.3 | 81.4 | **59.0** | **22.4** | 89.1 | 35.2 | 53.9 | 78.5 | 54.8 | 23.7 |
| CelebA | CLIP (ViT-B-32) | 80.1 | 72.7 | 7.4 | 85.2 | **81.5** | **3.7** | 79.6 | 71.3 | 8.3 | 84.8 | 80.5 | 4.3 |
| | CLIP (ViT-L-14) | 80.6 | 74.3 | 6.3 | 85.9 | **82.8** | 3.1 | 80.0 | 73.1 | 6.9 | 85.5 | 82.6 | **2.9** |
| | ALIGN | 81.8 | 77.2 | 4.6 | 83.9 | 78.0 | 5.7 | 83.9 | 81.4 | **2.5** | 86.3 | **83.4** | 2.9 |
| | AltCLIP | 82.3 | **79.7** | **2.6** | 86.1 | 75.6 | 10.5 | 81.9 | 79.0 | 2.9 | 86.0 | 77.2 | 8.8 |
| PACS | CLIP (ViT-B-32) | 96.7 | 82.1 | 14.6 | 97.0 | 83.7 | 13.3 | 96.6 | 84.2 | 12.4 | 97.0 | **86.3** | **10.7** |
| | CLIP (ViT-L-14) | 98.1 | 79.8 | 18.3 | 98.0 | 79.8 | 18.2 | 98.1 | 83.8 | 14.3 | 98.1 | **83.9** | **14.2** |
| | ALIGN | 95.8 | 77.1 | 18.7 | 95.8 | **78.0** | **17.8** | 95.1 | 71.1 | 24.0 | 95.0 | 73.8 | 21.2 |
| | AltCLIP | 98.5 | 82.6 | 15.9 | 98.4 | 83.0 | 15.4 | 98.6 | 88.8 | 9.8 | 98.7 | **89.5** | **9.2** |
| VLCS | CLIP (ViT-B-32) | 75.6 | 20.5 | 55.1 | 75.6 | 22.7 | 52.9 | 76.4 | 29.5 | 46.9 | 76.5 | **33.0** | **43.5** |
| | CLIP (ViT-L-14) | 72.6 | 4.2 | 68.4 | 70.9 | 6.8 | 64.1 | 73.4 | 8.9 | 64.5 | 71.1 | **12.6** | **58.5** |
| | ALIGN | 78.8 | 33.0 | 45.8 | 78.2 | 30.7 | 47.5 | 78.0 | **43.2** | **34.8** | 77.6 | 39.8 | 37.8 |
| | AltCLIP | 78.3 | 24.7 | **53.6** | 77.5 | 24.4 | 53.1 | 79.0 | 20.5 | 58.5 | 78.9 | **25.0** | 53.9 |
| CXR14 | BiomedCLIP | 55.3 | 28.9 | 26.4 | 55.7 | **41.8** | **13.9** | 54.8 | 21.8 | 33.0 | 56.2 | 41.6 | 14.6 |

We note that when doing both projections does not improve the baseline, using only $u^k$ or $v^j$ still outperforms the baseline. For instance, the ALIGN model in the Waterbirds dataset achieves the best performance with only $u^k$ projection. This suggests that in certain cases, harmful and helpful concepts are intertwined in the embedding space, and using just one projection can be beneficial. We leave further investigation to future work.

## F    ADDITIONAL EXPERIMENTS

### F.1    COMBINATION WITH THE CALIBRATION METHODS

Table 14: Additional baseline: text-classification calibration method Holtzman et al. (2021)

| Dataset | Model | Calibration | | | ROBOSHOT | | | Calibration + ROBOSHOT | | |
|---|---|---|---|---|---|---|---|---|---|---|
| | | AVG | WG(↑) | Gap(↓) | AVG | WG(↑) | Gap(↓) | AVG | WG(↑) | Gap(↓) |
| CivilComments | BERT | 51.0 | 37.3 | 13.7 | 49.7 | **42.3** | **7.4** | 53.4 | 36.9 | 16.5 |
| | Ada | 73.3 | 31.2 | 42.1 | 56.6 | **44.9** | **11.7** | 68.3 | 35.0 | 33.3 |
| HateXplain | BERT | 60.9 | 15.8 | 45.1 | 57.3 | 14.0 | 43.3 | 56.7 | **22.8** | **33.9** |
| | Ada | 61.9 | 31.6 | 30.3 | 63.6 | 21.1 | 42.5 | 59.6 | **33.3** | **26.3** |
| Amazon | BERT | 78.0 | 57.7 | 20.3 | 81.0 | **64.4** | **16.6** | 79.0 | 59.2 | 19.8 |
| | Ada | 71.2 | 50.5 | 20.7 | 82.9 | 63.8 | **19.1** | 83.2 | **63.9** | 19.3 |
| Gender Bias | BERT | 85.4 | 83.2 | 2.2 | 85.1 | **84.9** | **0.2** | 85.7 | 82.5 | 3.2 |
| | Ada | 84.2 | 77.8 | 6.4 | 78.0 | 60.1 | 17.9 | 84.2 | **77.9** | **6.3** |

Table 14 shows that ROBOSHOT further benefits from the calibration methods. This further highlights the versatility of ROBOSHOT—we can combine it with such methods with no additional work. To showcase this, we show additional results from (1) applying the calibration method alone, (2) our method, (3) the combination.

This result show that the best performing method across the board is either ROBOSHOT or the combination. The underlying reason for this is that as the two methods are orthogonal, adding calibration can further improve the results.

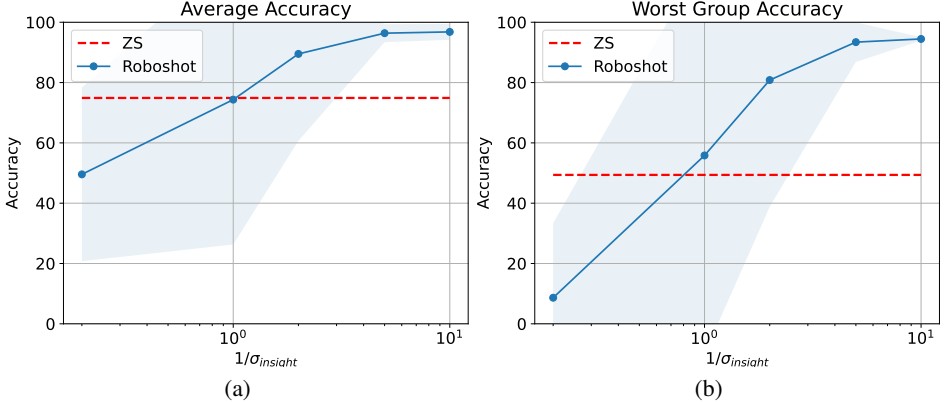

Figure 3: Synthetic experiment with varying $\sigma_{noise}$. As expected, the performance improves at a rate inversely proportional to $\sigma_{noise}$.

## F.2 SYNTHETIC EXPERIMENTS

**Setup.** We validate our theoretical claims by performing a synthetic experiment where we vary the noise level in the insight vectors ($\sigma_{insight}$). Higher $\sigma_{insight}$ indicates more noise. We use the following basis vectors as concept vectors $z_{helpful} = (1,0,0), z_{spurious} = (0,1,0), z_{benign} = (0,0,1)$, and class embedding vectors $c_1 = z_{helpful} + z_{spurious} + z_{benign}$ and $c_0 = -z_{helpful} - z_{spurious} + z_{benign}$. Experiments are repeated 100 times.

- Synthetic data input distribution ($s$ denotes spurious feature group)
  - $x|y = 1, s = 0 \sim \mathcal{N}([w_{helpful}, w_{spurious}, w_{benign}], \sigma_{input}I), n = 2500$
  - $x|y = 1, s = 1 \sim \mathcal{N}([w_{helpful}, -w_{spurious}, w_{benign}], \sigma_{input}I), n = 2500$
  - $x|y = 0, s = 0 \sim \mathcal{N}([-w_{helpful}, -w_{spurious}, w_{benign}], \sigma_{input}I), n = 2500$
  - $x|y = 0, s = 1 \sim \mathcal{N}([-w_{helpful}, w_{spurious}, w_{benign}], \sigma_{input}I), n = 2500$

- Insight vectors
  - $v_{helpful} = \gamma_{helpful}z_{helpful} + \gamma_s z_{spurious} + \gamma_b z_{benign}$, where $\gamma_s \sim \mathcal{N}(0, \sigma_{inisght})$, $\gamma_b \sim \mathcal{N}(0, \sigma_{benign})$
  - $v_{harmful} = \gamma_c z_{helpful} + \gamma_{harmful}z_{spurious} + \gamma_b z_{benign}$, where $\gamma_c \sim \mathcal{N}(0, \sigma_{inisght})$, $\gamma_b \sim \mathcal{N}(0, \sigma_{benign})$

For the experiment reported in Figure 3, we used $w_{helpful} = 1, w_{spurious} = 1, w_{benign} = 0.5, \gamma_{helpful} = 1, \gamma_{harmful} = 1, \sigma_{input} = 0.5, \sigma_{benign} = 0.01$

**Results.** In Figure 3, we observe that up to 10 - 20% of noise level to signal (harmful, helpful coefficients = 1), our algorithm works well, recovering worst group accuracy and improving average group accuracy. This result supports our claims in Theorems 4.1 and 4.2.

## F.3 EMBEDDING ANALYSIS

We provide insights into the case where our method does not improve the baseline (ALIGN model on Waterbirds) in Fig. 4. In Fig. 4a, we visualize the original and projected input embeddings ($x$ in green and red points, respectively), and the label embeddings ($c^0$ and $c^1$). Fig. 4a (left) shows the embeddings from the ALIGN model. We observe that the projected embeddings (red) still lie within the original embedding space, even with reduced variance. In contrast, when examining the CLIP model embeddings (Figure 4a (right)), we observe that the projected embeddings are significantly distant from the original ones. Unsurprisingly, Figure 4b (left) reveals that $v^j$ and $u^k$ (harmful and helpful insight embeddings in black and blue stars, respectively) are not distinguishable in the text embedding space of ALIGN, collapsing the input embeddings after ROBOSHOT is applied.

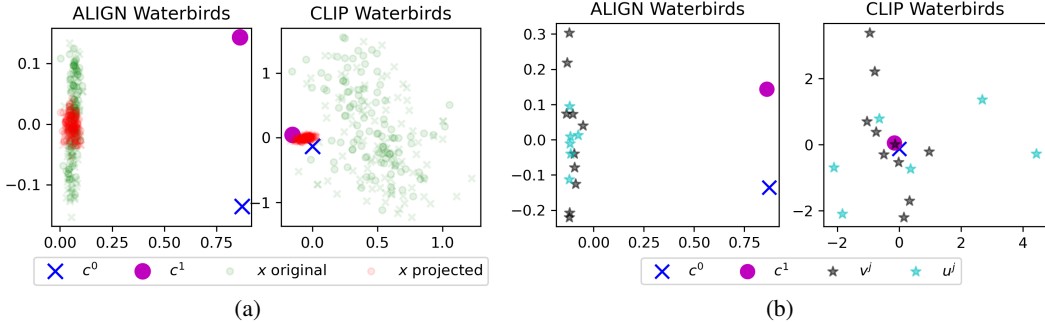

(a)                                        (b)

Figure 4: (a) Original (green) and projected (red) input embeddings $x$, and label embeddings $c^0$ and $c^1$. (b) label embeddings $c^0$ and $c^1$, harmful insight embeddings $v^k$ (black star) and helpful insight embeddings $u^j$ (blue star)

## F.4  ANALYSIS ON THE ROBUSTNESS TO SPURIOUS CORRELATIONS.

We provide in-depth result analysis to explain the performance changes in the average accuracy (AVG) and worst group accuracy (WG), especially with respect to spurious correlations. Concretely, consider the distribution of the margin $M : \mathcal{X} \to \mathbb{R}$ given by $M(x) := \langle c^+, x \rangle - \langle c^-, x \rangle$, where $c^+, c^-$ are the correct/incorrect class embeddings. Accuracy can be expressed as $\mathbb{EI}(M(x))$. The margin distributions and the margin changes by roboshot are illustrated in Figure 5 (Waterbirds), 6 (CelebA). We denote data with spurious features as $\mathcal{D}_{sp}$ (i.e. waterbirds with land background, landbirds with water background), and data with non-spurious features as $\mathcal{D}_{nsp}$ (i.e. waterbirds with water background, landbirds with land background). In the first column, $M(x)$ denotes the margin distribution of zeroshot prediction. In the second column, $M(\hat{x}_{rm}) - M(x)$ represents the margin changes by the roboshot harmful concept removal procedure. In the third column, $M(\hat{x}_{ad}) - M(\hat{x}_{rm})$ represents the margin changes by the roboshot helpful concept addition. Typically, inputs with spurious features $\mathcal{D}_{sp}$ tend to be closer to the decision boundary, inducing more errors. As expected, we can observe that harmful insight removal procedure increases the margin of $\mathcal{D}_{sp}$, but decreases the margin of inputs with non-spurious features $\mathcal{D}_{nsp}$. This can explain the potential tradeoff between the accuracy of $\mathcal{D}_{sp}$ and $\mathcal{D}_{nsp}$. If the gain in $\mathcal{D}_{sp}$ outweights the loss in $\mathcal{D}_{nsp}$, the average accuracy increases as in most cases. However, if the gain in $\mathcal{D}_{sp}$ is less the loss in $\mathcal{D}_{nsp}$, the average accuracy decreases as in ALIGN. In either case, the model performance in $\mathcal{D}_{sp}$ is improved by this procedure. In addition step, we expect that margins improve in both of $D_{sp}, D_{nsp}$ on average. Helpful insight addition procedure turns out be quite effective in CelebA dataset, where visual features can be described more easily by language models.

## F.5  ISOLATING CONCEPTS BY AVERAGING RELEVANT CONCEPTS

Table 15: Left: Cosine similarity between concept images and original embedding vs. averaged embedding. Right: ROBOSHOT on Waterbirds with original vs. averaged embedding

| Concept | Original | Average |
|---------|----------|---------|
| Green   | 0.237    | **0.241** |
| Red     | 0.236    | **0.240** |
| Blue    | 0.213    | **0.229** |
| Yellow  | 0.237    | **0.246** |
| Square  | 0.214    | **0.220** |

| | ZS | | | ROBOSHOT Original | | | ROBOSHOT Average | | |
|---|---|---|---|---|---|---|---|---|---|
| | AVG | WG | Gap | AVG | WG | Gap | AVG | WG | Gap |
| | 86.6 | 29.6 | 57.0 | 87.1 | 31.5 | 55.6 | 78.8 | **55.1** | **23.7** |

We conduct experiments to test the viability of our concept modeling. Specifically, we want to find out if CLIP input representation $x$ contains harmful, helpful, and benign components ($z_s$, $z_r$, and $z_b$ respectively in equation 1) and whether it is reasonable to assume benign components as noise.

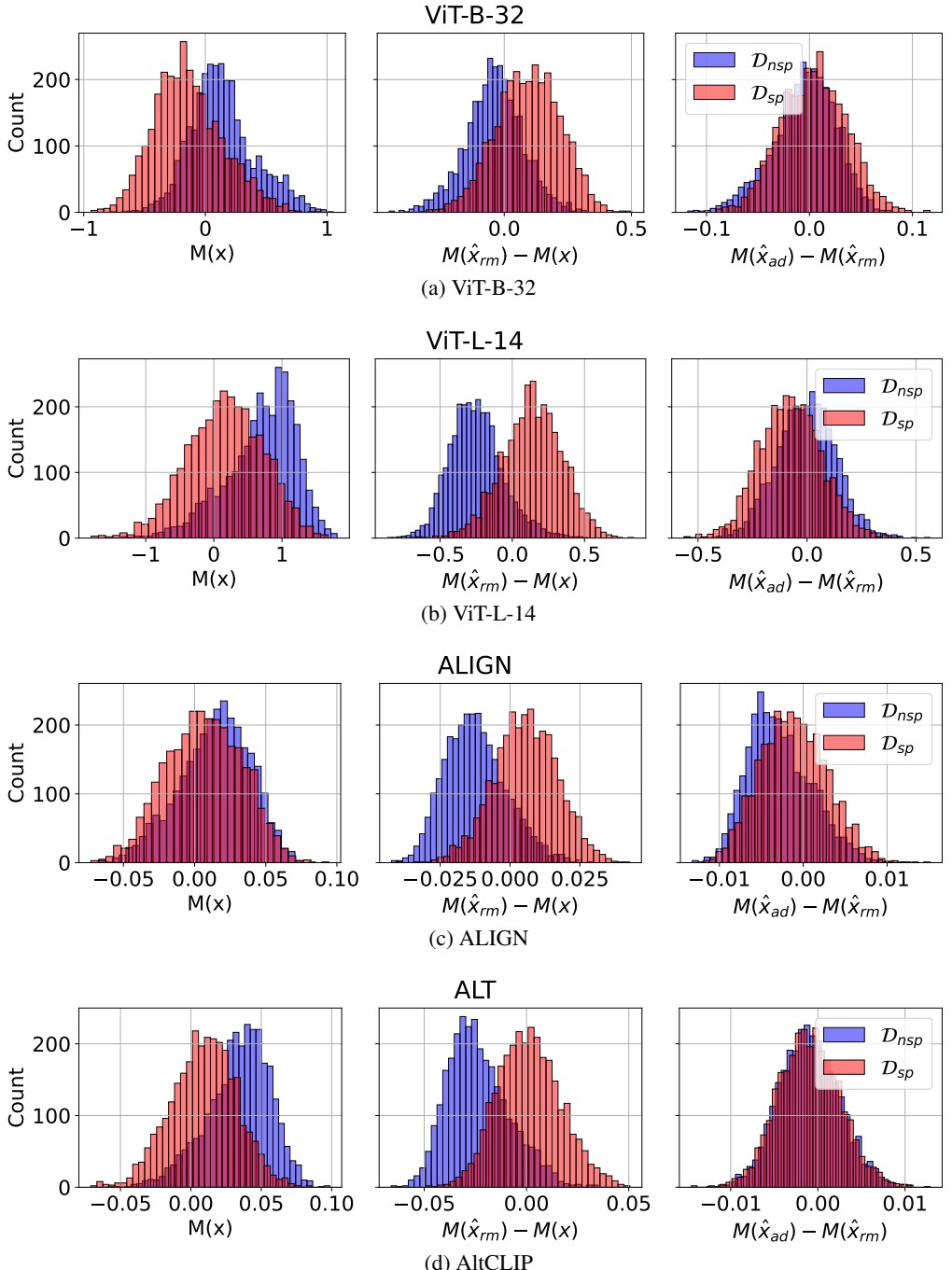

Figure 5: Margin analysis in Waterbirds dataset. Typically, inputs with spurious features $\mathcal{D}_{sp}$ tend to be closer to the decision boundary, inducing more errors. As expected, we can observe that harmful insight removal procedure increases the margin of $\mathcal{D}_{sp}$, but decreases the margin of inputs with non-spurious features $\mathcal{D}_{nsp}$. This can explain the potential tradeoff between the accuracy of $\mathcal{D}_{sp}$ and $\mathcal{D}_{nsp}$. If the gain in $\mathcal{D}_{sp}$ outweights the loss in $\mathcal{D}_{nsp}$, the average accuracy increases as in most cases. However, if the gain in $\mathcal{D}_{sp}$ is less the loss in $\mathcal{D}_{nsp}$, the average accuracy decreases as in ALIGN. In either case, the model performance in $\mathcal{D}_{sp}$ is improved by this procedure. In addition step, we expect that margin improves in both of $D_{sp}$, $D_{nsp}$ on average as in ViT-B-32. However, in most cases, the margin changes are not that crucial, implying extracting helpful insights is not easy in Waterbirds dataset.

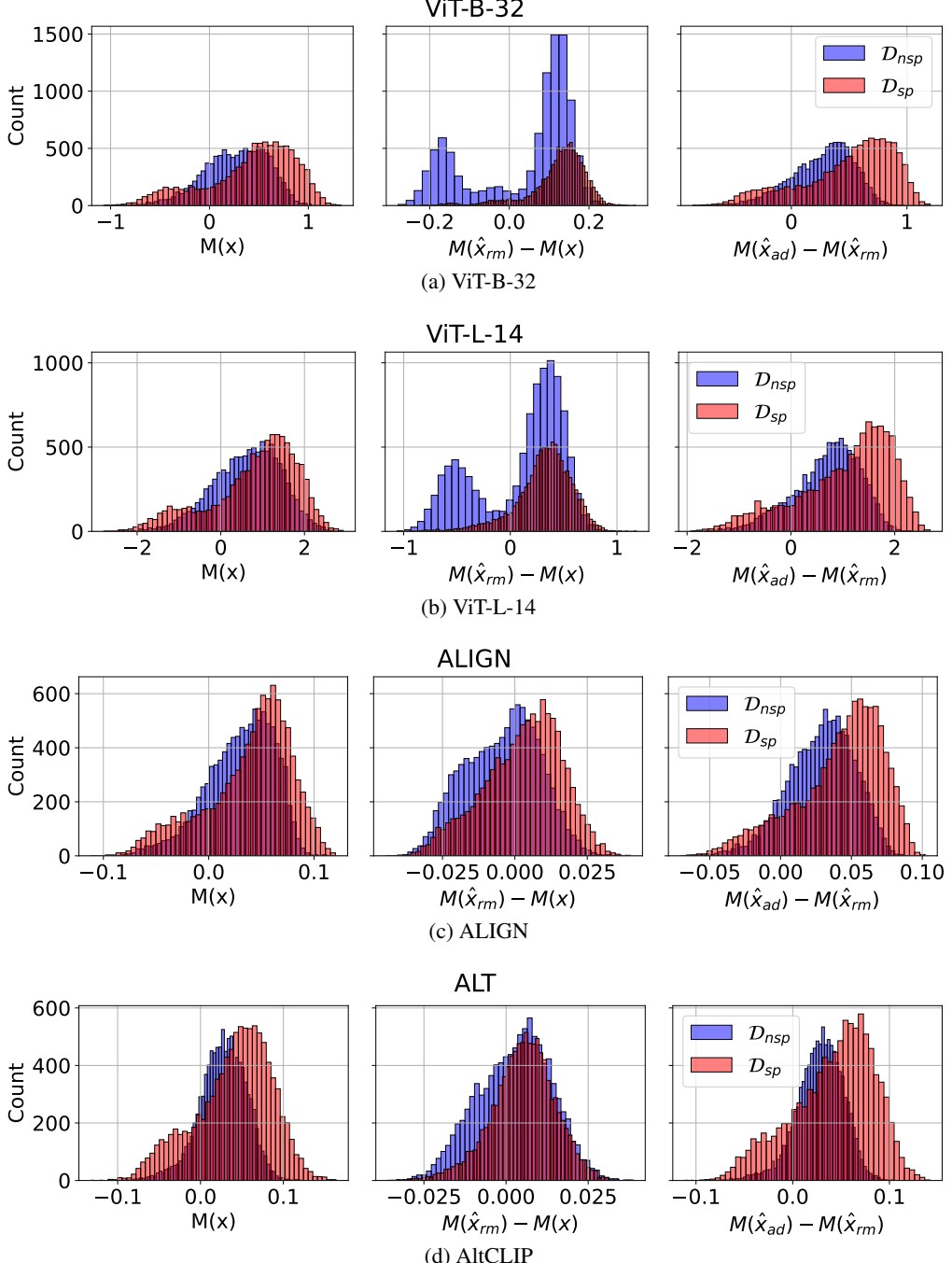

Figure 6: Margin analysis in CelebA dataset. Again, inputs with spurious features "blond" tend to induce errors ("men"-"blond", "girl"-"non-blond"). As expected, we can observe that harmful insight removal procedure increases the margin of $\mathcal{D}_{sp}$, but decreases the margin of inputs with non-spurious features $\mathcal{D}_{nsp}$, which may lead to the potential tradeoff. However, in CelebA dataset, the helpful insight addition step turns out to be helpful, increasing the margins of both distributions much. It can be interpreted as helpful insights can be captured easily in images.

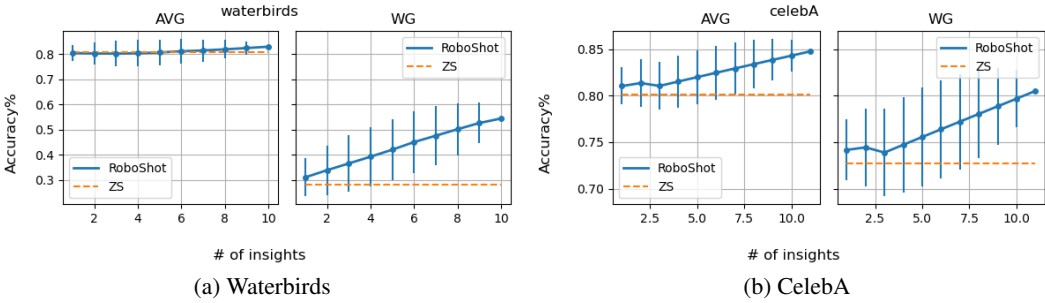

Figure 7: Number of insights ablations

**Can we partition CLIP input representation into harmful, helpful, and benign concepts?**  For a particular concept (e.g., "land"), we hypothesize that the true concept component is mixed with other concept components due to the signal in training data. For instance, land often co-occurs with sky, cattle, and other objects. Thus, the CLIP representation of "land" is entangled with these other concepts. To potentially isolate the helpful concept, we ask LM for an exhaustive list of concepts related to "land" and average the embedding of all related concepts. The intuition here is that a clean "land" component exists in each individual embedding, and the remaining is likely to be random, which can be averaged out and leave us with the true concept.

To verify this intuition, we compare the original and averaged embeddings of concepts listed in Table 15 (left). For each concept, we get 100 Google image search results and filter out noisy images (e.g., images with large text and artifacts) by eyeballing. We then report the average cosine similarity between the images and original embedding vs. the embedding from our averaging procedure. Averaged embedding has higher cosine similarity across the board than original CLIP embedding. To some extent, this indicates that the averaging procedure isolates the true concept. And thus, *benign components in embeddings can be canceled out.*

**Does ROBOSHOT gain improvement with isolated concept?**  Table 15 (right) compares ROBOSHOT with removing harmful insights using original CLIP embedding vs. averaged embedding. We use Waterbirds dataset because the harmful insights are known in prior. To isolate the effect of our averaging procedure, we use "landbird" and "waterbird" as labels without additional prompts (e.g., "a picture of [label]"), and we only use "land" and "water" as the harmful insights to remove, which causes slight difference with the results reported in Table 1. Confirming our intuition, *using the averaged embedding results in better WG performance and smaller Gap.*

### F.6  ROBOSHOT WITHOUT DECOMPOSITION

To see the effectiveness of QR decomposition of insight vectors, we conduct additional ablation experiment of decomposition method. In Table 16, w/o QR ($v^j$ only), w/o QR ($u^k$ only), and w/o QR (both) represents roboshot rejection only, addition only, both without QR decomposition step. Contrary to our expectation, in binary classification (Waterbirds, CelebA), Roboshot method works well without QR decomposition. This can be interpreted as insights from LLM provide almost orthogonal vectors. However, in multiclass classification, where rejection, addition vectors are generated by combinatorially paring insights for each class, Roboshot method get worse. Especially, addition step collapse. While rejection step wears off the subspace that the insight vectors span and there couldn't be more difference, addition steps can push multiple times to the similar directions. From this ablation experiment, the benefits of obtaining subspace via decomposition can be explained by two ways. First, in removal step, it provides a clean way to remove the subspace that spurious features span. Secondly, int addition step, it prevents overemphasis on some helpful insight directions.

Table 16: Ablation of QR decomposition

| Dataset | Model | Roboshot w/ QR | | | w/o QR ($v^j$ only) | | | w/o QR ($u^k$ only) | | | w/o QR (both) | | |
|---|---|---|---|---|---|---|---|---|---|---|---|---|---|
| | | AVG | WG(↑) | Gap(↓) | AVG | WG(↑) | Gap(↓) | AVG | WG(↑) | Gap(↓) | AVG | WG(↑) | Gap(↓) |
| Waterbirds | CLIP (ViT-B-32) | 83.0 | 54.4 | 28.6 | 79.5 | 58.3 | 21.2 | 83.0 | 31.2 | 51.8 | 79.6 | **62.5** | **17.1** |
| | CLIP (ViT-L-14) | 79.9 | 45.2 | 34.7 | 79.3 | 36.3 | 43.0 | 88.8 | 31.6 | 57.2 | 75.0 | **45.8** | **29.2** |
| | ALIGN | 50.9 | 41.0 | 9.9 | 53.3 | 36.6 | 16.7 | 62.0 | **50.9** | 11.1 | 38.2 | 36.5 | **1.7** |
| | AltCLIP | 78.5 | 54.8 | 23.7 | 70.8 | **56.1** | 14.7 | 89.0 | 35.0 | 54.0 | 64.3 | 52.8 | **11.5** |
| CelebA | CLIP (ViT-B-32) | 84.8 | 80.5 | 4.3 | 85.3 | 81.6 | 3.7 | 80.5 | 73.2 | 7.3 | 86.5 | **83.5** | **3.0** |
| | CLIP (ViT-L-14) | 85.5 | **82.6** | **2.9** | 86.1 | 81.7 | 4.4 | 79.7 | 72.5 | 7.2 | 85.8 | 80.0 | 5.8 |
| | ALIGN | 86.3 | 83.4 | 2.9 | 84.4 | 78.9 | 5.5 | 83.9 | 81.5 | 2.4 | 86.8 | **84.5** | **2.3** |
| | AltCLIP | 86.0 | 77.2 | 8.8 | 86.5 | 75.6 | 9.9 | 80.4 | 75.6 | **4.8** | 86.0 | **77.8** | 8.2 |
| PACS | CLIP (ViT-B-32) | 97.0 | **86.3** | **10.7** | 97.0 | 82.9 | 14.1 | 85.5 | 37.8 | 47.7 | 83.8 | 33.0 | 50.8 |
| | CLIP (ViT-L-14) | 98.1 | **83.9** | **14.2** | 98.0 | 79.8 | 18.2 | 84.9 | 13.4 | 71.5 | 85.8 | 11.8 | 74.0 |
| | ALIGN | 95.0 | 73.8 | 21.2 | 95.7 | **75.9** | **19.8** | 56.9 | 0.2 | 56.7 | 58.0 | 0.2 | 57.8 |
| | AltCLIP | 98.7 | **89.5** | **9.2** | 98.4 | 83.1 | 15.3 | 67.8 | 4.0 | 63.8 | 65.0 | 2.8 | 62.2 |
| VLCS | CLIP (ViT-B-32) | 75.6 | **33.0** | 43.5 | 75.5 | 20.5 | 55.0 | 21.4 | 0.0 | **21.4** | 30.7 | 0.0 | 30.7 |
| | CLIP (ViT-L-14) | 71.1 | **12.6** | 58.5 | 71.1 | 6.9 | 64.2 | 22.3 | 0.0 | 22.3 | 22.1 | 1.3 | **20.8** |
| | ALIGN | 77.6 | **39.8** | 37.8 | 78.1 | 33.0 | 45.1 | 36.2 | 0.0 | 36.2 | 32.7 | 0.1 | **32.6** |
| | AltCLIP | 78.9 | 25.0 | 53.9 | 77.5 | **25.1** | 52.4 | 31.4 | 0.0 | 31.4 | 30.6 | 2.0 | **28.6** |
| CXR14 | BiomedCLIP | 56.2 | **41.6** | **14.6** | 55.9 | 36.6 | 19.3 | 55.2 | 23.9 | 31.3 | 56.1 | 37.2 | 18.9 |

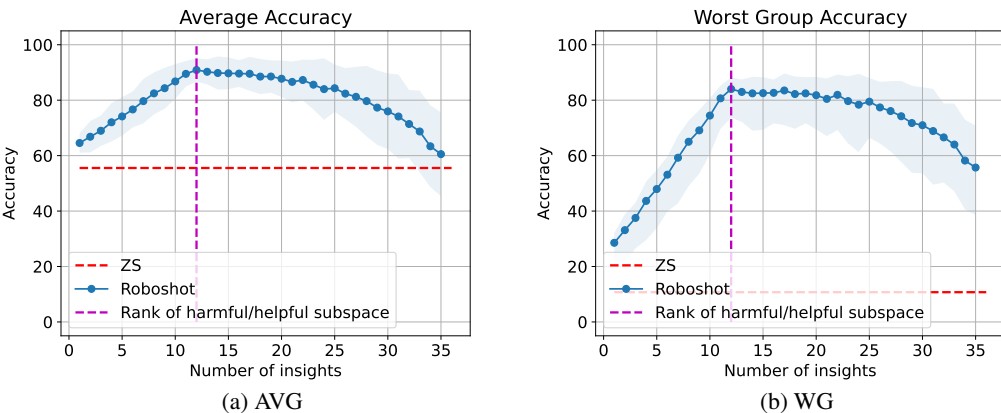

(a) AVG       (b) WG

Figure 8: Synthetic experiment on the number of insights

### F.7 INSIGHTS ANALYSIS

In Figure 7 we show ablations on the number of insights used in ROBOSHOT. We can clearly observe that both average and worst-group accuracy improves (almost) linearly with the number of insights used.

In our theory, increasing the number of insights is beneficial until the span of insight vectors cover the subspace of helpful/harmful features — thus for the optimality, we need insight vectors up to the rank of helpful/harmful subspaces. To validate it, we conduct a synthetic experiment with varying the number of insights. Here, the number of helpful/harmful/benign concepts is 12 respectively, with the embedding dimension 36 (12 + 12 +12). We add sequentially increase the number of insights based on a similar synthetic experiment setup in Appendix F.2 — after all target concepts are covered by at least one insight, we resample insights for each target concept. In Figure 8, we can observe that the AVG/WG performance improves until the number of insights is the same with the rank of harmful/helpful subspace. More detailed synthetic experiment setup is as follows.

- $z_s = e_s$, where $S = 12$ and $1 \leq s \leq S$

- $z_r = e_r$, where $R = 12$ and $S + 1 \leq r \leq S + R$
- $z_b = e_b$, where $B = 12$ and $S + R + 1 \leq b \leq S + R + B$
- $v_{spurious} = \sum_{s=1}^{S} z_s$
- $v_{helpful} = \sum_{r=S+1}^{S+R} z_r$
- $v_{benign} = \sum_{b=S+R+1}^{S+R+B} z_b$
- Synthetic data input distribution ($s$ denotes spurious feature group)
    - $x|y = 1, s = 0 \sim \mathcal{N}([w_{spurious}v_{spurious}, w_{helpful}v_{helpful}, w_{benign}v_{benign}], \sigma_{input}I), n = 2500$
    - $x|y = 1, s = 1 \sim \mathcal{N}([-w_{spurious}v_{spurious}, w_{helpful}v_{helpful}, w_{benign}v_{benign}], \sigma_{input}I), n = 2500$
    - $x|y = 0, s = 0 \sim \mathcal{N}([-w_{spurious}v_{spurious}, -w_{helpful}v_{helpful}, w_{benign}v_{benign}], \sigma_{input}I), n = 2500$
    - $x|y = 0, s = 1 \sim \mathcal{N}([w_{spurious}v_{spurious}, -w_{helpful}v_{helpful}, w_{benign}v_{benign}], \sigma_{input}I), n = 2500$
- Insight vectors follow the same assumptions in theory — the only target concept has the constant coefficient ($\gamma_{i,i}$ for $1 \leq i \leq S + R$) and other coefficients are sampled from $\mathcal{N}(0, \sigma_{insight})$ or $\mathcal{N}(0, \sigma_{benign})$.
- For the experiment reported in Figure 8, we used $w_{helpful} = 1, w_{spurious} = 0.5, w_{benign} = 0.01, \gamma_{i,i} = 1$ for $1 \leq i \leq S + R, \sigma_{input} = 1, \sigma_{benign} = 1$

