# OpenReview forum: "Zero-Shot Robustification of Zero-Shot Models"
_ICLR.cc/2024/Conference — ICLR 2024 poster_

### Official Review · Reviewer_1hyF · 2023-10-28

**Soundness:** 3 good
**Presentation:** 3 good
**Contribution:** 3 good
**Rating:** 6
**Confidence:** 4

**Summary:**

This paper proposed a zero-shot method to improve the robustness of pre-trained model embeddings. The key idea is to leverage insights obtained from language models based on task descriptions. After extracting the insights, they use the insights to modify the image embeddings, removing harmful components and enhancing useful ones, without any supervision. To achieve this goal, the method encourages invariant representation (to spurious features) by projecting the pre-trained model embeddings onto the subspace orthogonal to the subspace spanned by spurious feature descriptions. Experiments demonstrate that the proposed method improves multi-modal and language model zero-shot performance.

**Strengths:**

- **Novel and useful setting:** The setting of improving the robustness of pretrained model embeddings with task description is novel. RoboShot offers a unique approach that preserves the out-of-the-box usability of pretrained models, which is a key advantage.

- **Extensive experiments and analyses:** The authors demonstrated the efficacy of the proposed method and setting with extensive experiments and analyses, in terms of both datasets and settings.

- The paper is well-written.

**Weaknesses:**

The robustification relies on the insights provided by language models. However, if the language model does not identify the potential failure cases of the model, the method cannot remedy it. For instance, if the LM does not propose background as a spurious feature, can the method still mitigate such spurious correlation?

**Questions:**

N/A

---

> ### Author Response · Authors · 2023-11-18
>
> Thank you for noting **the novelty of our method**, and the strength of our evaluations!
> * **On scenarios where language models fail**. There are two cases to examine.
>
>     First, the reviewer asks about scenarios where the language model is not familiar with what causes, for example, CLIP, to fail. This is not a problem for our model: we do not need the LM to be aware of the intricacies and impacts of spurious correlations on a particular model. We only ask for relevant information about a task, not about a particular model's behavior on the task.
>
>     Second, there are scenarios where the language model cannot provide any task-specific insights. In this case, our method cannot directly produce any improvements, because we do not have anything to use for determining what the harmful subspace should be. *However, even in this setting, all hope is not lost:* There are ways to **transfer insights** from one task to another. This requires a form of meta-prompting: we ask language models for what tasks are similar to the given tasks, and then ask for insights about all of these similar tasks. For example, suppose that the language model provided no insights whatsoever for waterbird and landbird categorization. We could simply ask the language model for what type of task (and task entities) are similar. We may receive an answer about distinguishing mammals from lizards as a potentially similar task. In this case, background would be a transferable insight, since mammal/lizard categorization is similarly not affected by background. There is a large space for such approaches.

---

> ### Author Response · Authors · 2023-11-21
>
> Dear Reviewer,
>
> We thank you again for your feedback, questions, and suggestions! We believe we have answered all of your questions in our responses and the updated draft. If you have additional questions, we would love to answer them!
>
> The Authors

---

### Official Review · Reviewer_R1Zx · 2023-10-31

**Soundness:** 2 fair
**Presentation:** 3 good
**Contribution:** 3 good
**Rating:** 8
**Confidence:** 3

**Summary:**

The work tackles a very interesting and impactful topic, namely the improvement of zero-shot models without any additional labelled data, training or manual intervention. This goal is accomplished by leveraging existing pretrained Language Models (LMs) to infer positive and negative insigths from the task description, using their embeddings to obtain helpful, harmful and neutral subspaces and finally editing the representations to remove the harmful components while boosting the helpful ones. The paper then presents a theoretical analysis to characterize the conditions under which the framework allows correcting a wrong prediction. Finally, the framework is evaluated in a wide set of experiments, showcasing its benefits on worst-group accuracy when plugged on top of a varied set of baselines.

**Strengths:**

### Originality

- The paper proposes novel methodology to improve the robustness of zero-shot models such as CLIP without fine-tuning or using extra data.

### Quality

- The simplicity of the debiasing techniques makes it a cheap solution that can be easily employed by any practitioner.
- Accepting the assumptions, the theoretical analysis makes intuitive sense.

### Clarity

- The paper makes for a pleasing read: it is well written, easy to read and mostly clear.
- The framework is clearly explained in detail with a straightforward formalization and algorithmic outline, making it easy to reproduce. Source code is also provided.

### Significance

- The method is employed on a varied set of baselines (CLIP ViT-B-32, CLIP ViT-L-14, ALIGN, AltCLIP for zero-shot image classification and BERT and ADA for zero-shot text classification) assessing its general applicability
- The experimental evidence covers multiple datasets, namely Waterbirds, CelebA, PACS, VLCS and CXR14 for zs image classification and CivilComments, HateXplain, Amazon and Gender Bias for zs text classification. The datasets cover different domains.

**Weaknesses:**

- The assumption of concept embeddings being orthonormal doesn’t seem well motivated; do we expect the concept embedding of ‘waterbird’ to be orthogonal to ‘water’? I find the experiment in Appendix F.5 to be inconclusive due to the simplicity of the considered concepts, and I can’t immediately understand why the average of the images having a higher cosine similarity should give any insight on the decomposition of the space in harmful, helpful and neutral subspaces.
    - Unfortunately, the overall motivation and analysis seems to lay on this assumption, making it a core criticity of the work.
- The 15.98% improvement claim in the abstract actually regards the increase in Worst Group accuracy and not the overall improvement which is probably not positive, I find it should be stated clearly to avoid misleading the reader.
- The qualitative assessment does not really immediately convey the effect of increasing $u^k$. Some quantitative metrics would help, e.g. class separability measure such as the ratio of the inter-class distance to the average intra-class distance $\frac{d_{\text{inter}}}{\frac{1}{2} (d_{\text{intra}{C_1}} + d{\text{intra}_{C_2}})}$
- From the presentation perspective, the captions could be improved. Figure 1 could use a textual description to clarify what’s going on in the image, e.g. how does it go from having two projected embeddings and a single one in the right part.  Analogously, the caption of Figure 2 doesn’t state what are $Y_0$ and $Y_1$
- The framework fails in several cases to maintain the average accuracy of the baseline, being therefore only advisable when worst group accuracy is the metric of interest. Is this realistic? It would be nice to have a method that fell back to the standard setting when the approach proves to be detrimental.

**Questions:**

Table 1

- why does the model perform so much worse on Waterbirds except that for CLIP-B?

Figure 2

- what are Y_0 and Y_1? I guess they are some sort of class prototypes, but where are they defined?

---

> ### Author Response · Authors · 2023-11-18
>
> Thank you for noting the **novelty of our method**! We appreciate the kind comments and valuable feedback. We have make adjustments to figure captions as suggested by the reviewer in our updated draft.
>
> * **On the orthonormality assumption of concept embeddings**. We described these vectors as orthonormal purely for simplicity of explanation and derivation. However, **our method does not require the orthonormality assumption**. In general, if the $z_i$ are not orthonormal, we could describe them with a change of basis. The key requirement is that the subspaces where the helpful/harmful concepts live are not identical (i.e., as captured by the rank). Our method will fail if the subspaces of harmful concepts are, for example, identical to the subspace of helpful concepts. This is a weak assumption, however. Indeed, if it does not hold, harmful and helpful prediction concepts are totally aligned, so that good prediction is hopeless no matter what. We do not observe this to be the case in practice.
>
>     In terms of the underlying theoretical model, we offer two arguments in support of our position. First, the empirical measurements (given below). Second, similar findings have been proposed in a variety of recent work [1, 2, 3, 4, 5]:
> [1] Trager, Matthew, et al. "Linear spaces of meanings: compositional structures in vision-language models." Proceedings of the IEEE/CVF International Conference on Computer Vision. 2023.
> [2] Chuang, Ching-Yao, et al. "Debiasing vision-language models via biased prompts." arXiv preprint arXiv:2302.00070 (2023).
> [3] Park, Kiho, Yo Joong Choe, and Victor Veitch. "The Linear Representation Hypothesis and the Geometry of Large Language Models." arXiv preprint arXiv:2311.03658 (2023).
> [4] Jiang, Yibo, Bryon Aragam, and Victor Veitch. "Uncovering Meanings of Embeddings via Partial Orthogonality." Thirty-seventh Conference on Neural Information Processing Systems. 2023.
> [5] Dev, Sunipa, et al. "OSCaR: Orthogonal Subspace Correction and Rectification of Biases in Word Embeddings." Proceedings of the 2021 Conference on Empirical Methods in Natural Language Processing. 2021.
>
> * **On Appendix F5 results**. The purpose of this experiment is to establish further evidence for our theoretical model. We sample what we expect to be *cleaner* or less contaminated datapoints expressing concepts (e.g., an image of all yellow pixels, an image of a square). In other words, we expect that these datapoints are likely to have smaller coefficients for irrelevant concepts (and fewer relevant concepts). We hypothesize that  word embeddings with smaller harmful coefficient will be more similar in the embedding space with these *cleaner* images, and thus have a higher cosine similarity.
>
>     To address the reviewer's concern on concepts simplicity, we perform the same experiment using CIFAR10 test set.
>
> |Class|Original embedding|Average embedding|
> |-|:-:|:-:|
> |Horse|0.258|**0.261**|
> |Automobile|0.238|**0.251**|
> |Dog|0.237|**0.240**|
> |Bird|0.248|**0.260**|
> |Cat|0.232|**0.245**|
> |Airplane|0.241|**0.262**|
> |Ship|0.244|**0.253**|
> |Truck|0.243|**0.248**|
> |Deer|0.243|**0.269**|
> |Frog|0.235|**0.256**|
>
> Indeed, even on more complicated concepts like those in the CIFAR10 labels above, the text embeddings with noise component averaged out have higher cosine similarities to the corresponding *cleaner* image embeddings. This result increases our confidence in the theoretical model.
>
> *  **On the improvement claim.** Thank you for pointing this out! RoboShot produces average improvement of worst-group accuracy% of 15.95% with a trivial average decrease in average accuracy% of 1.44%. As suggested by the reviewer, have updated this number in the current abstract. We note that our findings are generally consistent with the robustness literature, where improving worst-group accuracy is the goal, and a decrease in accuracy overall is common. At the same time, we highlight that in several scenarios *both of thes metrices are improved.* We describe the reason for this in more depth below.

---

> > ### Author Response · Authors · 2023-11-18
> >
> > *  **On qualitative metrics.** Thank you for bringing this to our attention! As suggested by the reviewer, we measured the class separability metric ($\frac{d_{inter}}{.5(d_{intraC_1}+d_{intraC_2})}$) on the binary datasets (Waterbirds and CelebA). We take cosine similarity as the distance function. The results shows that our method increases class separability - further strengthening its efficacy.
> >
> > #### Model = CLIP-ViT-B-32 ####
> > |Dataset|Zero-Shot (baseline)|Ours ($v^j$ only)|Ours ($u^k$ only)|Ours (both)|
> > |-|:-:|:-:|:-:|:-:|
> > |Waterbirds|1.051|1.051|**1.053**|1.051|
> > |CelebA|1.003|1.005|1.047|**1.05**|
> > #### Model = CLIP-ViT-L-14 ####
> > |Dataset|Zero-Shot (baseline)|Ours ($v^j$ only)|Ours ($u^k$ only)|Ours (both)|
> > |-|:-:|:-:|:-:|:-:|
> > |Waterbirds|1.078|1.077|1.079|**1.08**|
> > |CelebA|1.01|1.012|1.06|**1.061**|
> > #### Model = ALIGN ####
> > |Dataset|Zero-Shot (baseline)|Ours ($v^j$ only)|Ours ($u^k$ only)|Ours (both)|
> > |-|:-:|:-:|:-:|:-:|
> > |Waterbirds|1.072|1.072|**1.079**|1.076|
> > |CelebA|1.003|0.999|**1.083**|1.059|
> > #### Model = AltCLIP ####
> > |Dataset|Zero-Shot (baseline)|Ours ($v^j$ only)|Ours ($u^k$ only)|Ours (both)|
> > |-|:-:|:-:|:-:|:-:|
> > |Waterbirds|**1.071**|1.067|1.069|1.066|
> > |CelebA|0.99|0.99|1.022|**1.023**|
> >
> >
> > In almost every case, harmful insights rejection ($v^j$ only) slightly decrease the class separation (as it decreases variance in the harmful direction); while helpful feature addition ($u^j$ only) increases the class separation (in most cases completely offseting the decrease caused by harmful feature rejection). By doing both, we obtain higher class separation than the original zero-shot classification.
> >
> > * **On average performance**. Indeed, the fact that average performance is decreased when improving worst-group accuracy is a known phenomenon, even when fine-tuning, e.g., see [1]. When removing harmful insights, RoboShot tries to remove spurious features which can be predictive for some groups (in-distribution data, e.g. water in waterbird images), but not across all groups (out-of-distribution data, e.g. water in landbird images). Thus, *accuracy in groups where spurious features were informative may drop slightly, while accuracy in groups where spurious features have adverse effects typically increases*. However, the tradeoff's appearance depends on the task, model, and embedding quality; **in many cases, average accuracy can substantially increase**. We note, as well, that this occurs only due to removal. When boosting helpful insights, our approach is beneficial to accuracy across all groups.
> >
> >     Thus, the case where average accuracy does not improve happens when 1) the removed harmful insights drop a group's accuracy in a way that outweighs the gains in other groups and 2) increasing helpful insights does not improve overall accuracy enough due to the embedding quality or weak helpful insights. Note that this tradeoff appears for the same reason even in fine-tuning based approaches, for example [1].
> >
> >     We also point the reviewer to Appendix F4, where we perform average vs. worst-group accuracy tradeoff in terms of the class margin.
> >
> >
> >     [1] Zhang, Michael, and Christopher Ré. "Contrastive adapters for foundation model group robustness.", NeurIPS 2022.
> >
> > * **On Waterbirds performance**. As pointed out by the reviewer, RoboShot improves worst-group accuracy in Waterbirds, with a bigger decrease in average accuracy than in other datasets. As described above we provide an explanation on this in Appendix F4, where we perform an analysis on average accuracy vs. worst-group accuracy trade-off from class margin perspective.
> >
> >     On Waterbirds, the margin gained by rare test points (e.g., waterbirds with land background) outweighs the margin lost by common test points (e.g., waterbirds with water background). This causes improvement in both worst-group and average accuracy---but occurrs only in CLIP-ViT-B-32. This is not the case in other models, which leads to average performance decrease.

---

> > ### Comment · Reviewer_R1Zx · 2023-11-20
> >
> > I thank the authors for their rebuttal. While I am still not totally convinced about concept embeddings being orthogonal, I understand that the method doesn't require them to be.
> > I find my concerns to be addressed by the response, and I am therefore raising my score to accept.

---

### Official Review · Reviewer_umVa · 2023-10-31

**Soundness:** 3 good
**Presentation:** 1 poor
**Contribution:** 4 excellent
**Rating:** 8
**Confidence:** 5

**Summary:**

The paper proposes to improve zeroshot performance of various foundation models by trying to segregate the representations into 3 set of orthonormal basis— harmful,helpful and benign vectors. Using a language model, the authors try to identify the set of harmful and helpful basis, and then try to remove/boost those basis accordingly. Overall, the problem is well motivated and gives good results. The method section however needs some efforts in writing to clarify some of the intricacies of the proposed approach.

**Strengths:**

- The overall motivation and idea of the paper is quite novel with interesting applications across various foundational models like CLIP and LLM.
- The empirical results also are quite strong.

**Weaknesses:**

- $X_{proj}$ has not been defined in LFA section.
- The authors should clarify how the basis vectors ($z$) are identified in the experiments, as the decomposition of insight vectors is based on that.
- I understand that the proposed approach is poised to give major gains in class imbalance settings or well known setting with spurious features. However, I encourage the authors to also provide results in standard classification tasks like imagenet using CLIP. I fear that in many standard tasks, removing these spurious features might hurt as well. However, one can always choose to not remove these.
- Do the authors have any insights or ablations as to how many insight vectors ($m$) is needed. I couldn’t see those details.

**Questions:**

See weakness section

---

> ### Author Response · Authors · 2023-11-18
>
> Thank your for pointing out the novelty of our method and the strength of our empirical results!
> * **On $X_{proj}$'s definition.** $X_{proj}$ is the sample embeddings post-application of the RoboShot procedure. We defined $X_{proj}$ in the parameters of Algorithm 2.
> * **On basis vectors ($z$).** The key strength of our method is that **it does not assume access to basis vectors $z$**. These vectors are part of our theoretical model only---but our approach does not require identifying them. In our model, we decompose embeddings as combinations of harmful, helpful, and benign components: $x = \sum_{s=1}^S \alpha_s^{\text{harmful}} z_s + \sum_{r=S+1}^{S+R} \alpha_r^{\text{helpful}} z_r + \sum_{b=S+R+1}^{S+R+B} \alpha_b^{\text{benign}} z_b.$
>
>     The goal of our procedure is to reduce $\alpha_s^{\text{harmful}}$ and increase $\alpha_{r}^{\text{helpful}}$ (the coefficient of harmful and helpful basis vectors). Since indeed these cannot be directly identified (i.e., we never know what $z_1$ is, for example), we query LLMs in the hope of text whose embeddings produce some information about these. Specifically, we obtain text from the LLM that we embedded. Such embeddings ideally span some subspace of the harmful $z$s that can be used to decrease $\alpha_s^{\text{harmful}}$ (and similary for the helpful concepts.). This does not ever require us to identify what $z_1, z_2, \ldots$ are.
>
> * **On standard classification tasks.** As suggested by the reviewer, we performed an experiment on ImageNet (using CLIP ViT-B-32). The performance of our method is roughly the same (we saw a ~1.1% difference). We do not expect RoboShot to produce improvement in this setting---we anticipate that many of the CLIP models have seen ImageNet or variants at training time and that no substantial biases are present that can be corrected. Fortunately, the result indicates that RoboShot largely retains performance in standard classification settings.
>
> * **On the number of insight vectors needed**. To address the reviewer's question, we do an ablation experiment and provide a theoretical characterization on number of insights needed. We provide the deatils in this response, and updated our manuscript with Appendix F7 that provides *empirical ablation results for Waterbirds an CelebA datasets*. As expected, we **observe that performance improves as the number of insights increases**.
>
>     First, we point out that individual insights have different qualities---further suggesting the need for multiple insight vectors. The folowing table shows the different performance results from using one individual insight on harmful vector rejection
>
> |Insight pair| AVG% | WG%|
> |-|:-:|:-:|
> ['a bird with coastal migration', 'a bird with inland migration']|76.0%|26.0%
> ['a bird with keratin feathers physiology', 'a bird with hydrophobic feathers physiology']|80.8%|29.3%|
> ['a bird that eats bugs.', 'a bird that eats mainly fish.']|79.6%|30.2%|
> ['a bird that lives in watery environments', 'a bird that lives on land.']|86.2%|52.2%|
>
> On the theoretical characterization: Increasing the number of insights is beneficial until the span of insight vectors becomes the subspace of helpful/harmful features. Thus, for optimality, we need insight vectors up to the rank of the helpful/harmful subspaces. A synthetic experiment where we vary the number of insights validates this hypothesis. Here, the number of helpful/harmful/benign concepts is 12 respectively, with the embedding dimension 36 (12 + 12 +12). We sequentially increase the number of insights based on a similar synthetic experiment setup in Appendix F.2. In Appendix F.7 Figure 8, We can observe that the AVG/WG performance improves until the number of insights is the same with the rank of harmful/helpful subspace. After this, due to the presence of noise in superfluous insights, helpful components are removed / harmful components are added, which results in a performance decrease.
>
>
> This suggests that *there is indeed a sweet spot for the number of insight vectors*. While our method works particularly well if each insight from the LLM exactly targets a helpful/harmful component, there can be multiple insights for each concept. To resolve this issue, our ongoing work includes the application of robust subspace learning, replacing the QR decomposition step. This step uses the entire subspace that insight vectors span. In constrast, robust subspace learning only use the subspace that has strong signals to mitigate the challenges induced by noise [1]. We are also developing a robust version of the theoretical characterization---but we leave this for future work.
>
> [1] Lerman, Gilad, and Tyler Maunu. "An overview of robust subspace recovery." Proceedings of the IEEE 106.8 (2018): 1380-1410.

---

> > ### Comment · Reviewer_umVa · 2023-11-21
> > **Thanks**
> >
> > Thanks for the clarifications.

---

### Official Review · Reviewer_bCze · 2023-11-06

**Soundness:** 3 good
**Presentation:** 3 good
**Contribution:** 3 good
**Rating:** 8
**Confidence:** 3

**Summary:**

The authors propose ROBOSHOT, a method for improving the robustness of pretrained models in zero-shot settings. It uses language models to obtain insights from task descriptions and uses these insights to remove harmful components and boost useful ones in model embeddings without any supervision.

The method is evaluated on nine image and NLP classification tasks and shows an average improvement of 15.98% over several zero-shot baselines. It is compatible with various pretrained and large language models and can further boost performance with a zero-shot label-free adaptation variant where there are a large number of unlabeled examples.

**Strengths:**

- The proposed ROBOSHOT method is an interesting novel approach that improves the robustness of zero-shot models against harmful concepts without the manual identification of harmful concepts. It leverages insights obtained from large language models to refine embeddings, and address inherited biases.  I also find the theoretical arguments interesting  which characterizes the conditions under which ROBOSHOT can outperform existing methods in zero shot learning.

- The  paper presents a well-structured and rigorous experimental evaluation across various datasets and model architectures.

- The paper is written in a clear and accessible manner, I like the detailed explanations of the ROBOSHOT algorithm, the theoretical framework, and the evaluation methodology.

- ROBOSHOT consistently outperforms several zero-shot baselines on multiple datasets. Having a powerful zero-shot learning method can address many real-life image classification tasks where labels are hard to come by.

**Weaknesses:**

- No large scale datasets like imagenet

- The benchmarks are limited to zero shot classification which is an easy task compared to zero-shot semantic segmentation and instance segmentation where this method could struggle.

- I don't see this work as actual zero shot because the pretrained model has so much information about the classes present in the chosen datasets. This work would be more impactful if the experiments were conducted on rare classes to test whether this method generalizes well.  ChatGPT has been trained on the internet, so this work is far from zero shot learning unless we include classes that are least likely to be seen by ChatGPT.

**Questions:**

Please address the weaknesses above.

---

> ### Author Response · Authors · 2023-11-18
>
> Thank you for noting the **novelty of our work** and the strong evaluation!
> * **On large scale datasets like ImageNet**. Our technique *does not require training*---so *large datasets are not a problem*. In general, the costs of our method are quite low for any dataset size. We obtain the insights and embed them, which is performed just once per dataset. These embeddings are re-used for modifying each datapoint's representation, which comes at only a marginal cost on top of inference. Indeed, some of the datasets in our experiments are reasonably large (e.g., CivilComments-WILDS has 133782 samples, Amazon-WILDS has 90078 samples). We list all dataset details in Appendix D1.
>
>     The reason we did not focus on ImageNet in particular is that we do not anticipate our method to produce any substantial improvements for this dataset. This is to be expected for two reasons: (i) CLIP-style models already perform well on ImageNet, and (ii) it is likely that ImageNet or variants are in the training data used for these models, so that biases or spurious correlations are not as harmful. We verified this by running our method on ImageNet, where we observed similar performance as with vanilla zero-shot classification.
>
> * **On zero-shot classification versus other tasks**. We anticipate that our technique is usable in such settings as well. In general, the goal of our technique is to remove some harmful aspects of pretrained models (and boost some helpful ones) --- this is done by the harmful feature rejection (Algorithm 1 line 3-4), and useful feature addition (Algorithm 1 line 7-8). The overall difficulty of the task is not the crucial ingredient, but rather the presence or absence of these components. For our ongoing work, we are investigating using our technique in concert with segmentation.
>
> * **On the zero-shot terminology**. We agree that there are a variety of uses of the term "zero-shot". The usage that we refer to is using a pretrained model **without** (1) modifying the prompt with additional examples (i.e., few-shot prompting), (2) fine-tuning a pretrained model with labeled data, (3) continual pretraining with additional domain-specific data. All of these methods are commonly used for task-specific predictions.
>
>     This usage is consistent with much of the literature. For example, CLIP's authors describe CLIP in the following way: "We ... turn CLIP into a zero-shot classifier."

---

> > ### Comment · Reviewer_bCze · 2023-12-05
> > **Response to Authors**
> >
> > Thanks for your rebuttal. I raised the score to 8.

---

> ### Author Response · Authors · 2023-11-21
>
> Dear Reviewer,
>
> We thank you again for your feedback, questions, and suggestions! We believe we have answered all of your questions in our responses and the updated draft. If you have additional questions, we would love to answer them!
>
> The Authors

---

### Meta-Review · Area_Chair_zeLq · 2023-12-07

**Metareview:**

This paper presents RoboShot, an innovative method that improves the robustness of pretrained model embeddings in a fully zero-shot fashion by leveraging insights from large language models to refine embeddings and address inherited biases. The theoretical framework provided characterizes the conditions under which RoboShot can outperform existing methods in zero-shot learning. The experimental evaluation is well-structured and rigorous, demonstrating consistent improvements over several zero-shot baselines across multiple datasets and model architectures. Additionally, the paper is clearly written and accessible, making it a valuable contribution to the field. Based on these strengths, I recommend acceptance of this paper.

**Justification For Why Not Higher Score:**

I think this is definitely a solild paper and a great contribution to the community. However, I think in terms of the methodology side, it doesn't sound super exciting (it can be similiar to a lot of methods using LMs recently). Therefore, I recommend the current score.

**Justification For Why Not Lower Score:**

n/a

---

### Decision · Program_Chairs · 2024-01-16

Accept (poster)